# ADDRESSING EXTRAPOLATION ERROR IN DEEP OFFLINE REINFORCEMENT LEARNING

## ABSTRACT

Reinforcement learning (RL) encompasses both online and offline regimes. Unlike its online counterpart, offline RL agents are trained using logged-data only, without interaction with the environment. Therefore, offline RL is a promising direction for real-world applications, such as healthcare, where repeated interaction with environments is prohibitive. However, since offline RL losses often involve evaluating state-action pairs not well-covered by training data, they can suffer due to the errors introduced when the function approximator attempts to extrapolate those pairs' value. These errors can be compounded by bootstrapping when the function approximator overestimates, leading the value function to *grow unbounded*, thereby crippling learning. In this paper, we introduce a three-part solution to combat extrapolation errors: (i) behavior value estimation, (ii) ranking regularization, and (iii) reparametrization of the value function. We provide ample empirical evidence on the effectiveness of our method, showing state of the art performance on the RL Unplugged (RLU) ATARI dataset. Furthermore, we introduce new datasets for bsuite as well as partially observable DeepMind Lab environments, on which our method outperforms state of the art offline RL algorithms.

## 1 INTRODUCTION

Agents are, fundamentally, entities which map observations to actions and can be trained with reinforcement learning (RL) in either an online or offline fashion. When trained online, an agent learns through trial and error by interacting with its environment. Online RL has had considerable success recently: on Atari (Mnih et al., 2015), the game of GO (Silver et al., 2017), video games like StarCraft II, and Dota 2, (Vinyals et al., 2019; Berner et al., 2019), and robotics (Andrychowicz et al., 2020). However, the requirement of extensive environmental interaction combined with a need for exploratory behavior makes these algorithms unsuitable and potentially unsafe for many real world applications. In contrast, in the offline setting (Fu et al., 2020; Fujimoto et al., 2018; Gulcehre et al., 2020; Levine et al., 2020), also known as batch RL (Ernst et al., 2005; Lange et al., 2012), agents learn from a fixed dataset which is assumed to have been logged by other (possibly unknown) agents. See also Fig. 1 for an illustration of these two settings. Learning purely from logged data allows these algorithms to be more widely applicable, including in problems such as healthcare and self-driving cars, where repeated interaction with the environment is costly and potentially unsafe or unethical, and where logged historical data is abundant. However these algorithms tend to behave considerably worse than their online counterpart.

Although similar in principle, there are some important differences between the two regimes. While it is useful for online agents to explore unknown regions of the state space so as to gain knowledge about the environment and better their chances of finding a good policy (Schmidhuber, 1991), this is not the case for the offline setting. Choosing actions not well-represented in the dataset for

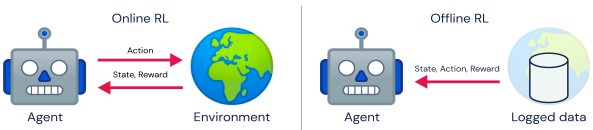

Figure 1: In online RL (left), the agent must interact with the environment to gather data to learn from. In offline RL (right), the agent must learn from a logged dataset.

offline methods would force the agent to rely on function approximators' extrapolation ability. This can lead to substantial errors during training, as well as during deployment of the agent. During training, the extrapolation errors are exacerbated by bootstrapping and the use of max operators (e.g. in Q-learning) where evaluating the loss entails taking the maximum over noisy and possibly overestimated values of the different possible actions. This can result in a propagation of the erroneous values, leading to extreme over-estimation of the value function and potentially unbounded error; see (Fujimoto et al., 2019b) and our remark in Appendix A. As we empirically show in Section 4.2, extrapolation errors are a different source of overestimation compared to those considered by standard methods such as Double DQN (Hasselt, 2010), and hence *cannot* be addressed by those approaches. In addition to extrapolation errors during training, a further degradation in performance can result from the use of greedy policies at test time which maximize over value estimates extrapolated to under-represented actions. We propose a coherent set of techniques that work well together to combat extrapolation error and overestimation:

**Behavior value estimation**. First, we address extrapolation errors during training time. Instead of $Q^{\pi^*}$, we estimate the value of the behavioral policy $Q^{\pi_B}$, thereby avoid the max-operator during training. To improve upon the behavioral policy, we conduct what amounts to a single step of policy improvement by employing a greedy policy at test time. Surprisingly, this technique with only one round of improvement allows us to perform significantly better than the behavioral policies and often outperform existing offline RL algorithms.

**Ranking regularization**. We introduce a max-margin based regularizer that encourages the value function, represented as a deep neural network, to rank actions present in the observed rewarding episodes higher than any other actions. Intuitively, this regularizer pushes down the value of all unobserved state-action pairs, thereby minimizing the chance of a greedy policy selecting actions under-represented in the dataset. Employing the regularizer during training will minimize the impact of the max-operator used by the greedy policy at test time, i.e. this approach addresses extrapolation errors both at training and (indirectly) at test time.

**Reparametrization of Q-values**. While behavior value estimation typically performs well, particularly when combined with ranking regularization, it only allows for one iteration of policy improvement. When more data is available, and hence we can trust our function approximator to capture more of the structure of the state space and as a result generalize better, we can rely on Q-learning which permits multiple policy improvement iterations. However this exacerbates the overestimation issue. We propose, in addition to the ranking loss, a simple reparametrization of the value function to disentangle the scale from the relative ranks of the actions. This reparametrization allows us to introduce a regularization term on the scale of the value function alone, which reduces over-estimation.

To evaluate our proposed method, we introduce new datasets based on bsuite environments (Osband et al., 2019), as well as the partially observable DeepMind Lab environments (Beattie et al., 2016). We further evaluate our method as well as baselines on the RL Unplugged (RLU) Atari dataset (Gulcehre et al., 2020). We achieve a new state of the art (SOTA) performance on the RLU Atari dataset as well as outperform existing SOTA offline RL methods on our newly introduced datasets. Last but not least, we provide careful ablations and analyses that provide insights into our proposed method as well as other existing offline RL algorithms.

**Related work**. Early examples of offline/batch RL include least-squares temporal difference methods (Bradtke & Barto, 1996; Lagoudakis & Parr, 2003) and fitted Q iteration (Ernst et al., 2005; Riedmiller, 2005). Recently, Agarwal et al. (2019a), Fujimoto et al. (2019b), Kumar et al. (2019), Siegel et al. (2020), Wang et al. (2020) and Ghasemipour et al. (2020) have proposed offline-RL algorithms and shown that they outperform off-the-shelf off-policy RL methods. There also exist methods explicitly addressing the issues stemming from extrapolation error (Fujimoto et al., 2019b).

## 2 BACKGROUND AND PROBLEM STATEMENT

We consider, in this work, Markov Decision Processes (MDPs) defined by $(\mathcal{S}, \mathcal{A}, P, R, \rho_0, \gamma)$ where $\mathcal{S}$ is the set of all possible states and $\mathcal{A}$ all possible actions. An agent starts in some state $s_0 \sim \rho_0(\cdot)$ where $\rho_0(\cdot)$ is a distribution over $\mathcal{S}$ and takes actions according to its policy $a \sim \pi(\cdot|s)$, $a \in \mathcal{A}$, when in state $s$. Then it observes a new state $s'$ and reward $r$ according to the transition distribution $P(s'|s, a)$ and reward function $r(s, a)$. The state action value function $Q^\pi$ describes the expected

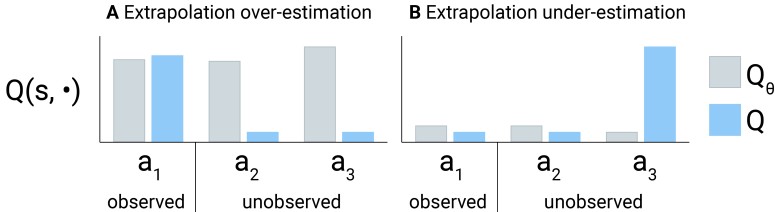

Figure 2: Two types of extrapolation error. Type A is most dangerous for offline RL, due to the max operation. Type B is difficult to address without additional interactions with the environment. Here, we aim to address Type A extrapolation errors.

discounted return starting from state $s$ and action $a$ and following $\pi$ afterwards:

$$Q^\pi(s,a) = \mathbb{E}\left[\sum_{t=0} \gamma^t r(s_t, a_t)\right], s_0 = s, a_0 = a, s_t \sim P(\cdot|s_{t-1}, a_{t-1}), a_t \sim \pi(\cdot|\boldsymbol{s}_t), \quad (1)$$

and $V^\pi(s) = \mathbb{E}_{a\sim\pi(\cdot|\boldsymbol{s})}Q^\pi(s,a)$ is the state value function. The optimal policy $\pi^*$, which we aim to discover through RL, is one that maximizes the expected cumulative discounted rewards, or *expected returns* such that $Q^{\pi^*}(s,a) \geq Q^\pi(s,a)\ \forall s,a,\pi$. For notational simplicity, we denote the policy used to generate an offline dataset as $\pi_\mathcal{B}$[1]. In the same vein, for a state $s$ in an offline dataset, we write $G^\mathcal{B}(s)$ to denote an empirical estimate of $V^{\pi_\mathcal{B}}(s)$, computed by summing future discounted rewards over the trajectory that $s$ is part of.

Approaches to RL can be broadly categorized as either *on-policy* or *off-policy* algorithms. Whereas on-policy algorithms update their current policy based on data generated by that same policy, off-policy approaches can take advantage of data generated by other policies. Algorithms in the mold of fitted Q-iteration make up many of the most popular approaches to deep off-policy RL (Mnih et al., 2015; Lillicrap et al., 2015; Haarnoja et al., 2018). This class of algorithms learns a $Q$ function by minimizing the Temporal Difference (TD) error. To increase stability and sample efficiency, the use of experience replay is also typically employed. For example, DQN (Mnih et al., 2015) minimizes the following loss function:

$$\mathcal{L}_{\theta'}(\theta) = \mathbb{E}_{(s,a,r,s')\sim\mathcal{D}}\Big(Q_\theta(s,a) - \big(r + \gamma \max_{a'} Q_{\theta'}(s',a')\big)\Big)^2, \quad (2)$$

where $\mathcal{D}$ represents experience replay, i.e. a dataset generated by some behavior policy. Typically, for off-policy algorithms the behavior policy is periodically updated to remain close to the policy being optimized. A deterministic policy can be derived by being greedy with respect to $\hat{Q}$, i.e. by defining $\pi(s) = \arg\max_a Q(s,a)$. In cases where maximization is nontrivial (e.g. continuous action spaces), we typically adopt a separate policy $\pi$ and optimize losses similar to: $\mathcal{L}_{\theta'}(\theta) = \mathbb{E}_{(s,a,r,s')\sim\mathcal{D}}\Big(Q_\theta(s,a) - \big(r + \gamma\mathbb{E}_{a'\sim\pi(\cdot|s')}[Q_{\theta'}(s',a')]\big)\Big)^2$. In this case, $\pi$ is optimized separately in order to maximize $\mathbb{E}_{a\sim\pi(\cdot|s)}[Q(s,a)]$, sometimes subject to other constraints (Lillicrap et al., 2015; Haarnoja et al., 2018). Various extensions have been proposed for this class of algorithms, including but not limited to: distributional critics (Bellemare et al., 2017), prioritized replays (Schaul et al., 2015), and n-step returns (Kapturowski et al., 2019; Barth-Maron et al., 2018; Hessel et al., 2017).

In the offline RL setting (see Figure 1, right), agents learn from fixed datasets generated via other processes, thus rendering off-policy RL algorithms particularly pertinent. Many existing offline RL algorithms adopt variants of Equation (2) to learn value functions; e.g. Agarwal et al. (2019b). Offline RL, however, is different from off-policy learning in the online setting. The dataset used is finite and fixed, and does not track the policy being learned. When a policy moves towards a part of the state space not covered by the behavior policy(s), for example, one cannot effectively learn the value function. We will explore this in more detail in the next subsection.

## 2.1 EXTRAPOLATION AND OVERESTIMATION IN OFFLINE RL

In the offline setting, when considering all possible actions for a next state in Equation (2), some of the actions will be *out-of-distribution* (OOD), i.e. these actions were never picked in that particular state by the behavior policy used to construct the training set (hence not present in the data). In such

---

[1]Our proposed approach does not depend on $\pi_\mathcal{B}$ being a coherent policy.

circumstances, we have to rely on the current Q-network's ability to extrapolate beyond the training data, resulting in extrapolation errors when evaluating the loss. Moreover, the need for extrapolation can lead to value overestimation, as explained below.

Value overestimation (see Fig. 2) happens when the function approximator predicts a larger value than the ground truth. In short, taking the max over actions of several Q-network predictions, as in Equation (2), leads to overconfident estimates of the true value of the state. We will expand on this point shortly. Before doing so, it is worth pointing out that this phenomenon of overestimation was well-studied in the online setting (Van Hasselt et al., 2015; 2018) and some prior works sought to address this problem (Van Hasselt et al., 2015; Fujimoto et al., 2018). However, in offline RL overestimation manifests itself in more problematic ways, which cannot be addressed by the solutions proposed in online RL (Kumar et al., 2019). To see this, let us consider Equation (2) again. The $\max$ operator is used to evaluate $Q_{\theta'}$ for all actions in a given state, including actions absent in the dataset (OOD actions). For OOD actions, we depend on extrapolated values provided by $\hat{Q}_{\theta'}$. While being an extremely powerful family of models, neural networks will produce erroneous predictions on unobserved state-action pairs, and sometimes, these will be artificially high. These errors will be propagated in the value of other states via bootstrapping. Due to the smoothness of neural networks, by increasing the value of actions in the OOD action-state's neighborhood, the overestimated value itself might increase, creating a vicious loop. Mainly we remark that, in such a scenario, typical gradient descent optimization can diverge and escape towards infinity. See Appendix A for a formal statement, and proof on this statement, though similar observations had been made before by Fujimoto et al. (2019b) and Achiam et al. (2019). In the online setting, when the agent overestimates some state-action pairs, they will be chosen more often due to optimistic estimates of values, even in the off-policy setting where the behavior policy trails the learned one. The online agent would then act, collect data, thereby correcting extrapolation errors. This form of self-correction is absent in the offline setting, and due to the overestimation from extrapolation, this absence can be catastrophic.

## 3 Solutions to Address Extrapolation Error

We build towards a solution to the extrapolation errors by i) using behavior value estimation to reduce training time extrapolation error, ii) ranking regularization of the Q-networks to better handle test time extrapolation error, iii) reparameterizing the Q-function to prevent divergence of these predictions to infinity.

### 3.1 Behavior Value Estimation

One potential answer to the overestimation problem is to remove the $\max$-operator in the policy evaluation step by optimizing the alternative loss:

$$\mathcal{L}_{\theta'}(\theta) = \mathbb{E}_{(s,a,r,s',a')\sim\mathcal{D}}\Big(Q_\theta(s,a) - \big(r + \gamma Q_{\theta'}(s',a')\big)\Big)^2. \tag{3}$$

This update rule relies on transitions $(s, a, r, s', a')$ collected by the behavior policy $\pi_\beta$ and resembles the policy evaluation step of SARSA (Rummery & Niranjan, 1994; Van Seijen et al., 2009). Since the update contains no $\max$-operator, and $Q_\theta$ are evaluated only on state-action pairs that are part of the dataset, the learning process is not affected by overestimation. However, the removal of the $\max$-operator means the update simply tries to evaluate the value of the behavioral policy. The astute reader may question our ability to improve upon the behavioral policy when using this update rule. We note, that when acting using the greedy policy $\pi(s) = \arg\max_a Q_\theta(s, a)$ we are in fact performing a single policy improvement step. Fortunately, this one step is typically sufficient for dramatic gains as we show in our experiments (see for example Fig. 9). This finding matches our understanding that policy iteration algorithms typically do not require more than a few steps to converge to the optimal policy (Lagoudakis & Parr, 2003; Sutton & Barto, 2018, Chapter 4.3).

### 3.2 Ranking Regularization

Policy evaluation with Eq. (3) effectively reduces overestimation during training. But it also avoids learning the $Q$ values of OOD actions. Due to the lack of learning, these values are likely erroneous, and many will err on the side of overestimation, thus harming the greedy policy. This is in contrast with the tabular case, where all OOD actions will have a default value of $0$.

To robustify the policy improvement step, a natural choice is to regularize the function approximator such that it behaves more predictable on unseen inputs. Forcing the neural network to output $0$ for OOD actions might require very non-smooth behavior of the network—hence we choose a less harsh regularizer that asks the model only to assign lower values to state-action pairs that have not been observed during learning. We formulate this as a ranking loss which follows a typical hinge-loss approximation (Chen et al., 2009; Burges et al., 2005) for ranking problems. Given a transition from the dataset $(s_t, a_t)$ this can be formulated as

$$\mathcal{C}(\theta) = \sum_{i=0, i \neq t}^{|\mathcal{A}|} \max\left(Q_\theta(s_t, a_i) - Q_\theta(s_t, a_t) + \nu, 0\right)^2. \quad (4)$$

While equation (4) does, in expectation, encourage lower ranks for OOD action, it can also have the adverse effect of promoting suboptimal behavior that is frequent in the dataset. This is because for any transition, proportional to its frequency in the dataset, the regularizer pushes the value of all but the selected action down, promoting a policy that picks the selected action regardless of its value. To minimize this effect, we weigh the regularization based on the value of the trajectory:

$$\mathcal{C}(\theta) = \exp\left(\left(G^{\mathcal{B}}(s) - \mathbb{E}_{s \sim \mathcal{D}}[G^{\mathcal{B}}(s)]\right)/\beta\right) \sum_{i=0, i \neq t}^{|\mathcal{A}|} \max\left(Q_\theta(s, a_i) - Q_\theta(s, a_t) + \nu, 0\right)^2, \quad (5)$$

where $\mathbb{E}_{s \sim \mathcal{D}}[G^{\mathcal{B}}(s)]$ is estimated by average over $G^{\mathcal{B}}(s)$ in mini-batches. In all our experiments, we fix $\nu$ to be $5e - 2$ and $\beta$ to be $2$. The new formulation of the loss ensures that particularly on trajectories performed well in the dataset, trajectories that are likely for policy learned using behavior evaluation, the OOD action rank lower than observed actions. We note that our rank loss, when viewed through the lens of on-policy online RL, can be related to ranking policy gradients (Lin & Zhou, 2020) or to (Su et al., 2020; Pohlen et al., 2018) who also used a hinge loss as a regularizer but with different goals from ours.

### 3.3 REPARAMETRIZATION OF Q-VALUES

The overestimation of state-action values can be severe in offline RL, and can escape towards infinity (see for example appendix A). While behavior value estimation can be an effective way for suppressing this overestimation, when one iteration of policy improvement is insufficient, one may want to bring back the max-operator and therefore the implicit policy improvement step of Q-learning. To better handle this scenario, we introduce a complimentary method to prevent severe over-estimation by bounding the values predicted by the critic via reparameterization. Specifically, we reparameterize the critic as $Q_\theta(s, a) = \alpha \hat{Q}_\theta(s, a)$ given a state- and action-independent scale parameter $\alpha$. This, in effect, disentangles the scale from the relative magnitude of values predicted, but also enables us to impose constraints on the scale parameter. To further stabilize the learning and reduce the variance of the estimations, we update $\alpha$ by stochastic gradient descent, but with larger minibatches and a smaller learning rate.

In our formulation, the "standardized" value $\hat{Q}_\theta(s, a) \in [-1, 1]$ is attained by using a tanh activation function. Note that the tanh activation has the side effect of reducing numerical resolution for representing extreme values (as the tanh will be in its saturated regime), minimizing the ability of the learning process to keep growing these values by bootstrapping on each other. We let $\alpha = \exp(\rho)$ such that $\alpha > 0$. Our parameterization thus ensures that Q-values are always bounded in absolute value by $\alpha$, i.e. n $Q(s, a) \in [-\alpha, \alpha]$. The equations below show how critic scaling can be adapted into the Q-learning objective:

$$\mathcal{L}_{\theta', \alpha'}(\theta, \alpha) = \mathbb{E}_{(s, a, r, s') \sim \mathcal{D}}\left(\alpha \hat{Q}_\theta(s, a) - \left(r + \gamma \alpha' \max_{a'} \hat{Q}_{\theta'}(s', a')\right)\right)^2. \quad (6)$$

The introduction of $\alpha$ allows us to conveniently regularize the scale of $Q$ values without disturbing the ranking between actions. More precisely, we introduce a regularization term on $\alpha$

$$\mathcal{C}(\alpha) = \mathbb{E}[\mathrm{softplus}(\alpha \hat{Q}_\theta(s, a) - G^{\mathcal{B}}(s))^2], \quad (7)$$

where $\mathcal{C}(\alpha)$ represents a soft-constraint requiring $Q$ values to stay close to the performance of the behavioral policy, and thereby prevent gross overestimation. In Eq. (7), we rely on the softplus function to constrain $\alpha$ only when $Q_\theta(s, a) > G^{\mathcal{B}}(s)$.

## 4 EXPERIMENTS

We investigate the performance of discrete offline RL algorithms on the three aforementioned open-source domains: Atari, DeepMind Lab, and `bsuite`. A question we are particularly interested in answering is: how does the lack of coverage of the state-action pairs affect the performance of each algorithm? In that context, we study each algorithms' robustness to dataset size (see Fig. 6), noise (see Fig. 3 and 7), and reward distribution (Fig. 9 in Appendix), as they all affect the datasets' coverage of the state and action space.

Because we explore various ablations of our proposed approach, discussed in Section 3, we used a specific acronym for each potential combination. According to our naming convention, `Q` is for Q-learning and `B` is for behavior value estimation as the underlying RL loss, `R` indicates the use of the ranking regularization and `r` the use of reparametrization. In that vein, `QRr` refers to Q-learning with ranking regularization and reparametrization and `BR` stands for behavior value estimation with ranking regularization (see Appendix B.1.) We note that, both our DQN and R2D2 experiments used Double Q-learning (Van Hasselt et al., 2015), but for our approach (and ablations of it) that rely on Q-learning, we used the vanilla Q-learning algorithm. More details for each experimental setup appear in Appendix D. We also provide more analysis and additional results in Appendix B.

We used an open-source Atari offline RL dataset, which is a part of RL Unplugged (Gulcehre et al., 2020) benchmark suite. We have created two new offline RL datasets for `bsuite` and DeepMind Lab, which we are going to opensource. The details of those datasets are provided in Appendix C.

### 4.1 BSUITE EXPERIMENTS

`bsuite` (Osband et al., 2019) is a proposed benchmark designed to highlight key aspects of agent scalability such as exploration, memory, credit assignment, etc. We have generated low-coverage offline RL datasets for *catch* and *cartpole* as described by Agarwal et al. (2019a) (see Appendix C.1 for details).

In Fig. 3, we compare the performance of `BRr` and `QRr` with four baselines: DDQN (Hasselt, 2010), CQL (Kumar et al., 2020), REM (Agarwal et al., 2019a) and BCQ (Fujimoto et al., 2018). We consider two tasks, each in five versions defined by the amount of injected noise. The noise is injected into transitions by replacing the actions from an agent with a random action with probability $\epsilon$.

On the harder dataset (*cartpole*), `BRr`, the proposed method, outperforms all other approaches showing the efficiency of our approach and its robustness to noise. Two other methods, `QRr` (proposed by us as an ablation to `BRr`) and CQL, also perform relatively well. The results for *catch* are similar, with the exception that BCQ also improves performance which re-emphasises the importance of restricting behavior to stay close to the observed data. We have additional results on *mountain_car*, where most algorithms behave well except DDQN (see Appendix D.4).

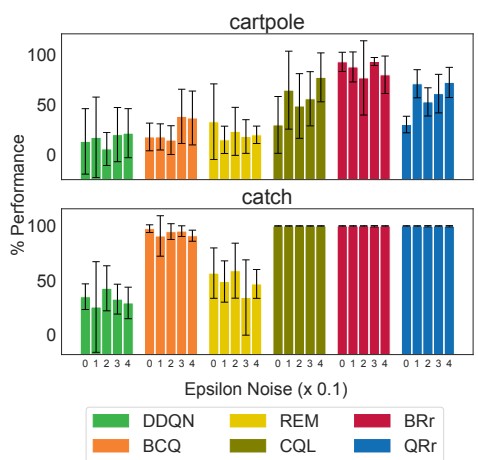

Figure 3: **Bsuite Experiments:** `bsuite` experimental results on two environments with respect to different levels of noise injected into the actions in the dataset. The proposed method, `BRr`, outperforms all the baselines on *cartpole*. Methods implementing a form of behavior constraining (BCQ, CQL and our methods `BRr` and `QRr`) excel on *catch*, stressing its importance.

### 4.2 ATARI EXPERIMENTS

Atari is an established online RL benchmark (Bellemare et al., 2013), which has recently attracted the attention of the offline RL community (Agarwal et al., 2019a; Fujimoto et al., 2019a) arguably because the diversity of games presents a challenge for offline RL methods. Here, we used the experimental protocol and datasets from the RL Unplugged Atari benchmark (Gulcehre et al., 2020). We report the median normalized score across the Atari games, and the error bars show a bootstrapped estimate of the $[25, 75]$ percentile interval for the median estimate computed across different games.

In Fig. 4, we show that `QRr` outperforms all baselines reported in the RL Unplugged benchmark as well as CQL (Kumar et al., 2020). While `BRr` performs well, this experiment highlights the potential limitation of doing a single policy improvement iteration in rich data regimes. Because in the considered setting there is enough data for the neural networks to learn reasonable approximations of the Q-value (exploiting the structure of the state space to extrapolate for unobserved state-action pairs), one can gain more by reverting to Q-learning in order to do multiple policy improvement steps. However this amplifies the role of the regularization and in particular the reparametrization. Therefore, in this setting, `QRr`, which we proposed as an ablation to `BRr` outperforms other techniques. Fig. 4 also shows the robustness of `QRr`'s hyperparameters to different tasks.

**Ablation Experiments on Atari** We ablate three different aspects of our algorithm on **online policy selection games**: i) the choice of TD backup updates (Q-learning or behavior value estimation), ii) the effect of ranking regularization, iii) the reparameterization on the critic. We show the ablation of those three components in Fig. 5. We observed the largest improvement when using ranking regularization. In general, we found that estimating the Monte-Carlo returns directly from the value function (we refer this in our plot as "MC Learning") does not work on Atari. However, behavior value estimation and Q-learning both have similar performance on the full dataset, however in low data regimes, the behavior value policy considerably outperforms Q-learning (see Fig. 9 in Appendix B).

**Overestimation Experiments** Q-learning can over-estimate due to the maximization bias, which happens due to the max-operator in the backups (Hasselt, 2010). In the offline setting another source of overestimation, as discussed in Section 2, are OOD actions due to the dataset's limited coverage. Double DQN (DDQN by Hasselt (2010)) is supposed to address the first problem, but it is unclear whether it can address the second. In Fig. 6, we show that in the offline setting DDQN still over-estimates severely when we evaluate the critic's predictions in the environment. We believe this is because the second factor is the main reason of overestimation, which is not explicitly addressed by DDQN. However, `Qr` (vanilla Q-learning with reparametrization) and `B` are not effected from the

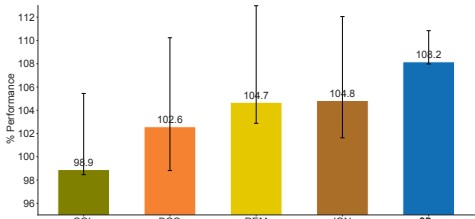

Figure 4: **Atari results:** We compare our proposed `QRr` results against other recent State of Art offline RL methods on the Atari **offline policy selection games** from RL Unplugged benchmark.

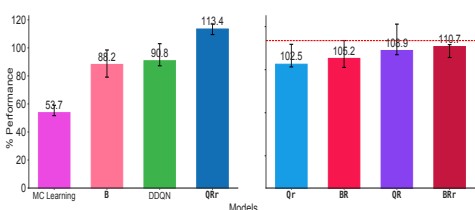

Figure 5: **Ablations (online policy selection games):** [LEFT] We compare behavior value estimation (`B`), DDQN and Monte Carlo approaches in offline RL Atari dataset. `B` and DDQN achieve similar median episodic returns, but learning with Monte Carlo returns performs poorly. [RIGHT] We show various ablation studies (in terms of using regularization, reparametrization and behavior value estimation). We found the most significant improvement from the ranking regularization term. Although the combination of ranking regularization and the reparameterization performs the best.

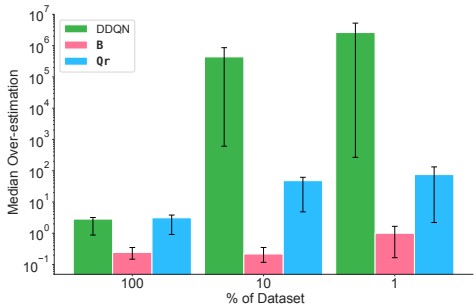

Figure 6: **Overestimation of Q-values** with sub-sampled Atari datasets (% of Dataset). DDQN over-estimates the value of states severely whereas `Qr` and `B` reduce over-estimation greatly. We report median over-estimation error over online policy selection games on Atari.

reduced dataset size and coverage as much. In the figure, we compute the over-estimation error as $\frac{1}{100} \sum_{i=0}^{100} (\max(Q^\pi(s,a) - G^\pi(s), 0))^2$ over 100 episodes, where $G^\pi(s)$ corresponds to the discounted sum of rewards from state $s$ till the end of episode by following the policy $\pi$.

**Robustness Experiments** In Appendix B.3 (see Figure 9), we investigate the robustness of `B` and DDQN with respect to the reward distribution and dataset sizes. We found that the performance of `B` is more robust than DDQN to the variations on the reward distribution and the dataset size.

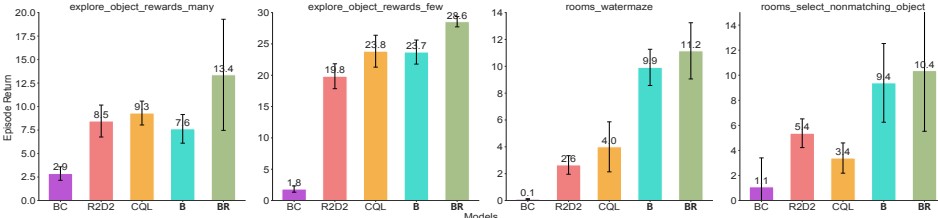

Figure 7: **DeepMind Lab Results:** We compare the performance of different baselines on challenging DeepMind Lab datasets coming from four different DeepMind Lab levels. Our method, `BR`, consistently performs the best.

### 4.3 DEEPMIND LAB EXPERIMENTS

Offline RL research mainly focused on fully observable environments such as Atari. However, in a complex partially observable environment such as Deepmind Lab, it is very difficult to obtain good coverage in the dataset even after collecting billions of transitions. To highlight this, we have generated datasets by training an online R2D2 agent on DeepMind Lab levels. Specifically, we have generated datasets for four of the levels: `explore_object_rewards_many`, `explore_object_rewards_few`, `rooms_watermaze`, and `rooms_select_nonmatching_object`. The details of the datasets are provided in the Appendix C.2.

We compare offline R2D2, CQL, BC, B and `BR` on our DeepMind Lab datasets. In contrast to Atari, `BR` performed better than `QR` according to our preliminary results. Thus, here, we decided to only focus on `BR`. We use the same network architecture and hence, the models vary only is the loss function. We want to compare our baselines' performance on our Deepmind Lab datasets when there is a large amount of data stored during online training with online R2D2. In Figure 7, we show the performance of each algorithm on different levels. Our proposed modifications, `BR` and `B` outperform other offline RL approaches on all DeepMind Lab levels. We argue that poor performance of R2D2 in the offline setting is due to the low coverage of the dataset. Despite having on the order 300M transitions, since the environment is partially observable and diverse, it is still not enough to cover enough of all possible state-action pairs. We present further results about dataset coverage on Deepmind Lab `seekavoid_arena_01` level with dataset generated by a fixed policy in Appendix B.2 where we showed that `BR` is more robust to the dataset coverage than other offline RL methods.

## 5 DISCUSSION

In this work, we first highlight how, in the offline deep RL setting, overestimation errors may cause Q-learning to diverge, with weights and Q-value escaping towards infinity. We discuss using *behavior value estimation* to address this problem, which efficiently regresses to the Q-value of the behavior policy and then takes a policy improvement step at test time by acting greedily with respect to the learnt Q-value. The behavior value estimation oversteps the overestimation issue by avoiding the max-operator during training. We note that a single policy improvement step seems sufficient, especially in the low data regime, to improve over the behavior policy and the policy discovered by double DQN. However, the max-operator used to construct the test time policy re-introduces overestimation errors that were avoided during training. We can address this issue by regularizing the function approximator with a ranking loss that encourages OOD actions to rank lower than the observed actions. This reduces overestimation at test time and improves performance. Nevertheless, we observe that *behavior value estimation* can be too conservative in rich data settings. In such scenarios, the function approximator can exploit more of the state and action space's underlying structure, leading to more reliable extrapolation. Therefore, it can be more lucrative to rely on Q-learning in such scenarios, which can do multiple policy improvement steps, further constraining the function approximator. The resulting algorithm `QRr`, that is Q-learning with the ranking loss and reparametrization, outperforms all other approaches on the RL Unplugged Atari benchmark.

Overall *behavior value estimation* coupled with the ranking loss, is an effective algorithm for low data regimes. For larger data regimes, where the coverage is better, it is possible to achieve better performance by switching to Q-learning and using reparametrization. The proposed methods outperform existing offline RL approaches on the considered benchmarks. As future work, we plan to extend our observations to the continuous control setup and towards more real-world applications.

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

## A    Q-LEARNING CAN ESCAPE TO INFINITY IN THE OFFLINE CASE

**Remark 1.** *Q-learning, using neural networks as a function approximator, can diverge in the offline RL setting given that the collected dataset does not include all possible state-actions pairs, even if it contains all transitions along optimal paths. Furthermore, the parameters (and hence the Q-values themselves) can espace towards infinity under gradient descent dynamics.*

*Proof.* The proof relies on providing a particular instance where Q-learning diverges towards infinity. This is *sufficient* to show that divergence can happen. Note that the remark does not make any statement of how likely is for this to happen, nor is providing sufficient conditions under which such divergence has to happen.

Let us consider a simple deterministic MDP depicted in the figure below (left).

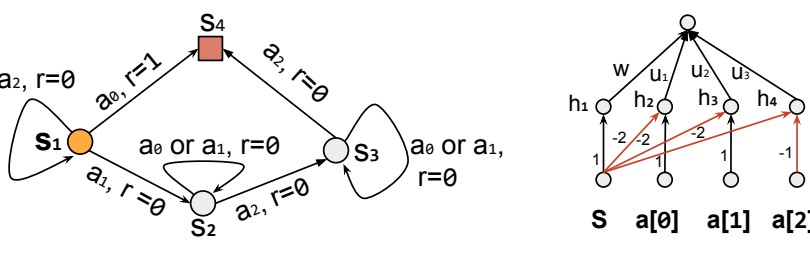

Depiction of the MDP              Depiction of the MLP

$\mathcal{S} = \{s_1, s_2, s_3, s_4\}$ is the set of all states, where $S_1$ is deterministically the starting state and $S_4$ is the terminal state of the MDP. Let $\mathcal{A} = \{a_0, a_1, a_2\}$ be the set of all possible actions. Let the reward function $r(s,a)$ be 0 for all action-state pair except $r(s_1, a_0)$ which is 1. Let the transition probabilities $P(s'|s, a)$ be deterministic as defined by the depicted arrows. I.e. for any state action pair only transitioning to one state has probability 1, while the rest has probability 0. For example, only $P(s_4|s_1, a_0) = 1$, while $P(s_3|s_1, a_0) = 0, P(s_2|s_1, a_0) = 0. P(s_2|s_1, a_0) = 0$. For $s_1, a_1$ only $P(s_2|s_1, a_1) = 1$ and so on and so forth.

*First observation* is that the optimal behavior is to pick action $a_0$ (as it is the only rewarding transition in the entire MDP).

The features describing each state are given by a single real number, where $s_1 = 0, s_2 = 1, s_3 = \beta$, with $\beta > \frac{1}{\gamma} > 0$, where $\gamma$ is the discount factor. Assume actions are provided to the neural network as one-hot vectors, i.e. $a_0 = [0, 0, 1]^T, a_1 = [0, 1, 0]^T, a_2 = [1, 0, 0]^{T\,2}$, where we will refer to $a[i]$ as the $i$-th element of the vector that represents the action $a$. For example $a_0[2] = 1$ and $a_0[0] = 0$.

Let us consider the Q-function parametrized as a simple MLP (depicted in the figure above left). The MLP uses rectifier activations, and gets as input both the state and action, returning a single scalar value which is the Q-value for that particular state action combination. Rewriting the diagram in analytical form we have that for $s \in \mathcal{R}$ and $a \in \mathcal{R}^3$:

$$Q_\theta(s, a) = w \cdot relu(s) + u_1 relu(a[0] - 2s) + u_2 relu(a[1] - 2s) + u_3 relu(-2s - a[2]) \quad (8)$$

*A note on initialization.* The weights of the first layer are given as constants. The process would work if we leave them to be learnable as well, but the analysis would become considerably harder. The exact value used, $-2, 1, -1$, are not important. In principle we care for the negative weights connecting $s$ to $h_2, h_3, h_4$ be larger in magnitude than those from $a[i]$ to $h_i$, and we care for the weight between $a[2]$ and $h_2$ to be negative. They can be scaled arbitrarily small and do not need to be identical.

---

[2]one-hot representation is the typical representation for action in discrete spaces

What we will rely in the rest of the analysis is that the preactivation of $h_2, h_3, h_4$ to be negative for state $s_2$ and $s_3$. This will be in the zero region of the rectifier, meaning no gradient will flow through those units. Since $s_3 > s_2 \geq 1$ and $a[i] \in \{0, 1\}$, it is sufficient for the weight from $s$ to $h_2, h_3, h_4$ to be larger in magnitude than the weight from $a[i]$ to $h_2, h_3, h_4$. This ensures that for $s > 1$, the Q-function is not a function of $u_i$ as $u_i$ will get multiplied by 0.[3] Also we want the function to never depend on $u_3$ to simplify our analysis, which is easily achievable if the weight going from $a[2]$ to $h_4$ is negative.

Given the observations above, if we plug in the formula the different values of $s_i$ and $a_i$ we get that:

$$
\begin{aligned}
& Q_\theta(s_1, a_0) && = u_1 \\
& Q_\theta(s, 1, a_1) && = u_2 \\
& Q_\theta(s_1, a_2) && = 0 \\
\forall a \in \mathcal{A}, \quad & Q_\theta(s_2, a) && = w \\
\forall a \in \mathcal{A}, \quad & Q_\theta(s_3, a) && = \beta w
\end{aligned}
\tag{9}
$$

Note that this implies that

$$
\begin{aligned}
\max_a Q_\theta(s_2, a) &= w \\
\max_a Q_\theta(s_3, a) &= \beta w
\end{aligned}
\tag{10}
$$

Assume $w > 0$. And let the dataset collected by the behavior policy to contain the following 3 transitions:

$$
\mathcal{D} = \{(s_1, a_0, 1, s_4), (s_1, a_1, 0, s_2), (s_2, a_2, 0, s_3)\}
$$

We can now construct the Q-learning loss that we will use to learn the function $Q$ in the offline case which will be

$$
\begin{aligned}
\mathcal{L} &= \sum_{(s,a,r,s') \in \mathcal{D}} (Q_\theta(s, a) - r - \gamma max_a Q'_\theta(s', a))^2 \\
&= (Q_\theta(s_1, s_0) - 1)^2 + (Q_\theta(s_1, a_1) - \gamma \max_a Q'_\theta(s_2, a))^2 + (Q_\theta(s_2, a_2) - \gamma \max_a Q'_\theta(s_3, a))^2 \\
&= (u_1 - 1)^2 + (u_2 - \gamma w')^2 + (w - \gamma \beta w')^2
\end{aligned}
\tag{11}
$$

Note that we relied on Eq. (10) to evaluate the $\max$ operator and $\theta'$ is a copy of $\theta$, that is used for bootstrapping. This is the standard definition of Q-learning see Eq. (2). In particular in this toy example $\theta'$ is numerically always identical to $\theta$ (in general it can be a trailing copy of $\theta$ from k steps back) and is used more to indicate that when we take a derivative of the loss with respect to $\theta$ we do not differentiate through $Q'_\theta$. From Eq. (11) we notice that only the first transition in dataset contributes to the gradient of $u_1$, only the second transition contributes to the gradient of $u_2$ and only the third transition contributes to the gradient of $w$. We can not evaluate the gradient with respect to $\theta$ of the loss $\mathcal{L}$ over the entire dataset:

$$
\begin{aligned}
\nabla_{u_1} &= u_1 - 1 \\
\nabla_{u_2} &= u_2 - (0 + \gamma w) \\
\nabla_{u_3} &= w - (0 + \gamma \beta w) && = (1 - \gamma \beta) w \\
\nabla_w &= w - (0 + \gamma \beta w) && = (1 - \gamma \beta) w
\end{aligned}
\tag{12}
$$

Note that we assumed $w > 0$ and for simplicity we exploited that $w' = w$ numerically, to be able to better understand the dynamics of the update. Given that $\beta > \frac{1}{\gamma}$, $\nabla_w$ will always be negative as long as $w$ (and implicitly $w'$) stays positive. Given that $w_t = w_{t-1} - \alpha \nabla_w$ for some learning rate $\alpha > 0$, the update creates a vicious loop that will increase the norm of $w$ at every iterations, such that $\lim_{t \to \infty} w_t = \infty$. Given that the gradient on $u_2$ tracks $w$, it means that the path that takes action $a_2$ in the initial state $s_1$ will have $+\infty$ as value. *Note that all transitions along the optimal path of this deterministic MDP are part of the dataset.*

---

[3]The fact that no gradient gets propagated in the first layer is only important if we attempt to consider the case when the first layer weights are learnable.

Also that given our example, the same will happen if we rely on SGD rather than batch GD (as the different examples affect different parameters of the model independently and there is no effect from averaging). Preconditioning the updates (as for e.g. is done by Adam or rmsprop) will also not change the result as they will not affect the sign of the gradient (the preconditioning matrix needs to be positive definite). Neither momentum will not affect the divergence of learning, as it will not affect the sign of the update.

This means that the provided MDP will diverge towards infinity under the updates on most commonly used gradient based algorithms.

□

## B  ADDITIONAL RESULTS AND ABLATIONS

### B.1  ACRONYMS

In Table 1, we provided the acronyms for our models and their corresponding meanings.

Table 1: Acronyms for our models and their expansions

| Acronym | Meaning |
| --- | --- |
| B | Behavior Value Estimation |
| BR | Behavior Value Estimation with Ranking Regularization |
| BRr | Behavior Value Estimation with Ranking Regularization and reparameterization |
| Q | Standard Q-learning |
| QR | Standard Q-learning with Ranking Regularization |
| QRr | Standard Q-learning with Ranking Regularization and reparameterization |

### B.2  DEEPMIND LAB: THE EFFECT OF COVERAGE ON OFFLINE LEARNING

As mentioned in Section 4.3 we investigate the effect of coverage on the DeepMind Lab seekavoid_arena_01 level. To do so, we have created another set of datasets which is generated by using a fixed R2D2 snapshot with different noise levels when evaluating the trained snapshot in the environment for seekavoid_arena_01 level. We have used different $\epsilon$s in the $\epsilon$-greedy algorithm to create datasets with different noise levels. The $\epsilon$ also effects the coverage of the dataset.

We compare R2D2, CQL, BC, B and BR on these DeepMind Lab datasets by using the same network architecture—the only change among the models is the loss function.

We investigate the effect of coverage in the Deep-Mind Lab seekavoid_arena_01 level, by evaluating the policy with different $\epsilon$'s for the epsilon-greedy in the environment and storing each episode in the dataset. Increasing the $\epsilon$ will increase the coverage of the dataset but also it will increase the noise in the dataset as well. In Figure 8, we show the effect $\epsilon$ on the simple DeepMind Lab level. When $\epsilon = 0$, BC works outperforms offline RL approaches, however increasing the level of noise deteriorates the performance of BC, and BR starts to performs better. Since the environment is deterministic, if the policy is deterministic as well, this corresponds to only having one single unique episode in the dataset. As we increase the epsilon the coverage in the dataset and diversity of the trajectories will increase as well. We trained all models by unrolling on the whole episode and trained using back-propagation through time.

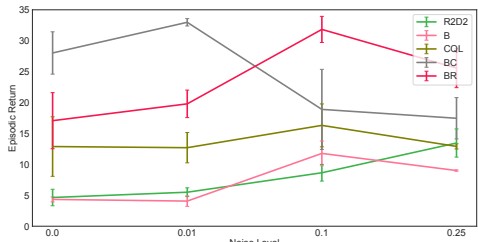

Figure 8: **Effect of coverage in the dataset:** We compare offline RL models with varying the noise level in the environment. Increasing the noise level increases the coverage as well. BC performs well with low noise, however, BR performs significantly better as the noise increases. Let us note that, in all our experiments, R2D2 uses double Q-learning.

## B.3    ATARI: ROBUSTNESS TO DATA

The robustness of the reward distribution in the dataset is an important feature required to deploy offline RL algorithms in the real-world. We would like to understand the robustness of behavior value estimation in the offline RL setting. Thus, we first investigate the robustness of B in contrast to Q-learning with respect to the datasets' size and the reward distribution. In Fig. 9, we split out the dataset into two smaller datasets: i) transitions coming from only highly rewarding ii) transitions from only poorly performing episodes. We show that B outperforms Q-learning in both settings.

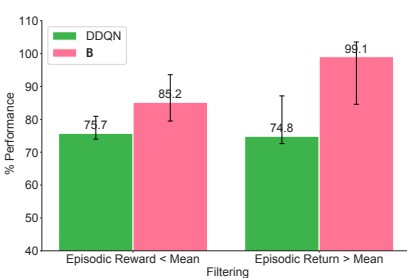
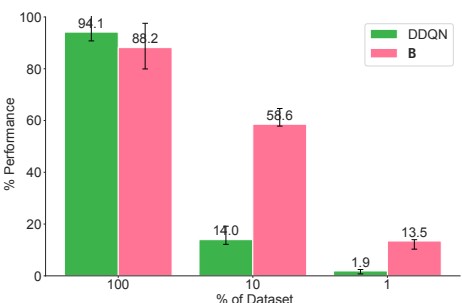

Figure 9: **Robustness Experiments:** (left) We compare DQN and B in terms of their robustness to the reward distribution on Atari online policy selection games. We split the datasets in two bins: the dataset that only contains transitions that are coming from episodes that have episodic return less than the mean episodic return in the dataset ("Episodic Reward < Mean"), transitions coming from episodes with return higher than the mean return in the dataset ("Episodic Reward > Mean"). B performs better than DQN in both cases. (right) Normalized scores of DQN and B on subsets of data from online policy selection games. B performs comparatively better than DQN. The Q-learning suffers more since the coverage of the dataset reduces with the subsampling which causes more severe extrapolation error.

## B.4    ON THE EFFECT OF REGULARIZATION

In this Section we study the effect of the regularization on the action gap and the overestimation error. In Figure 10, we show that increasing the regularization co-efficient for the ranking regularization increases the action gap across the Atari online policy selection games which can result to lower estimation error and better optimization.

In Figure 11, we show the effect of increasing the regularization on the overestimation of the Q-network when evaluated in the environment. We show the mean over-estimation across the games.

## B.5    ONLINE POLICY SELECTION GAMES RESULTS

In Figure 12, we show the performance of different models with respect to the rewards they achieve over the training.

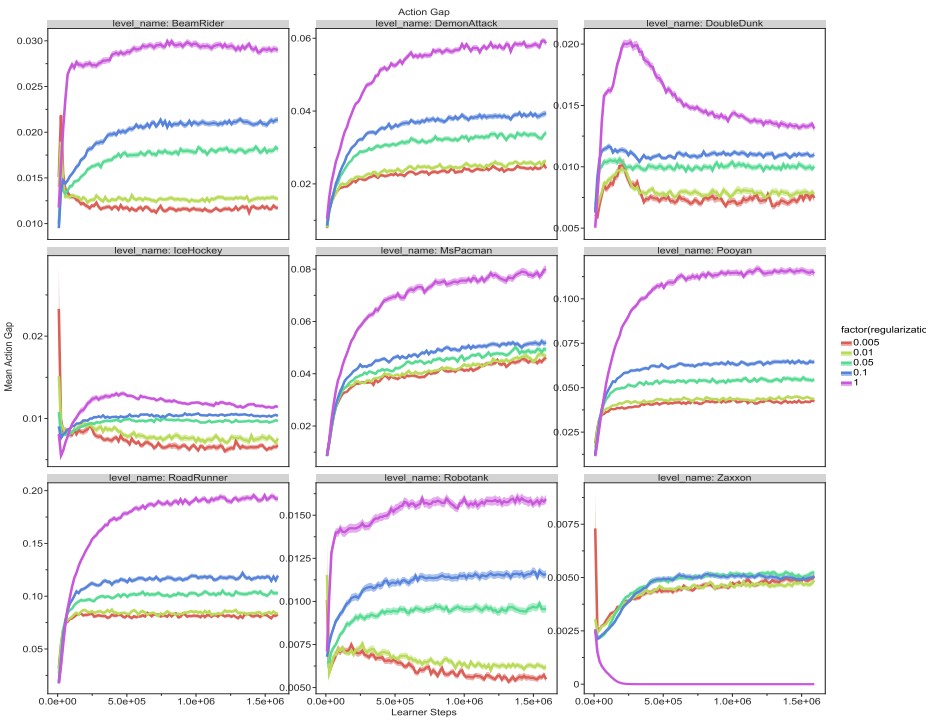

Figure 10: The Effect of increasing the ranking regularization on the action gap.

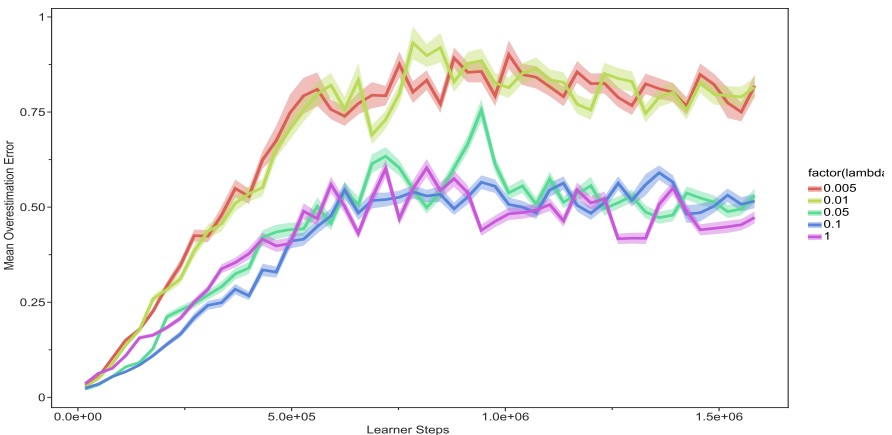

Figure 11: The Effect of increasing the ranking regularization on the overestimation.

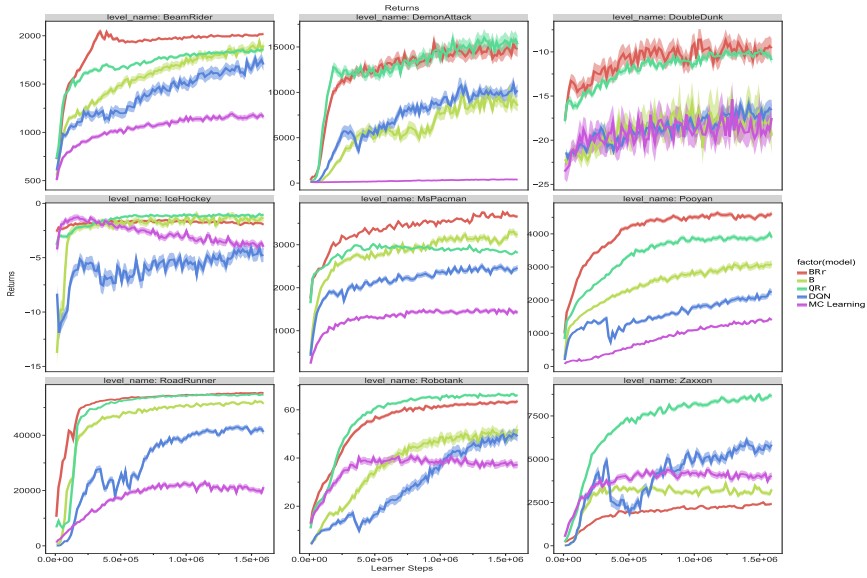

Figure 12: The Raw Returns obtained by each baseline on Atari online Policy Selection Games.

## B.6 OVERESTIMATION ON ONLINE POLICY SELECTION GAMES

In Figure 13 and 14, we report the value error of B, BRr and DQN's value error and squared value error respectively.

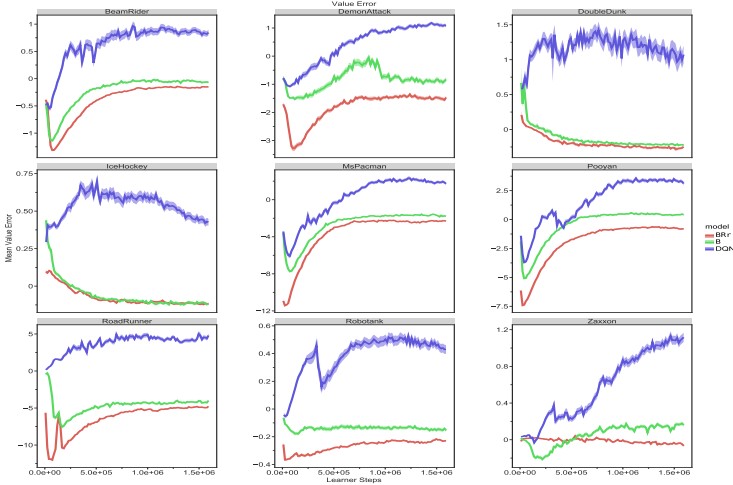

Figure 13: The value error computed in the environment by evaluating the agent and computed with respect to the ground truth discounted returns. The negative values indicate under-estimation and positive values are for over-estimation.

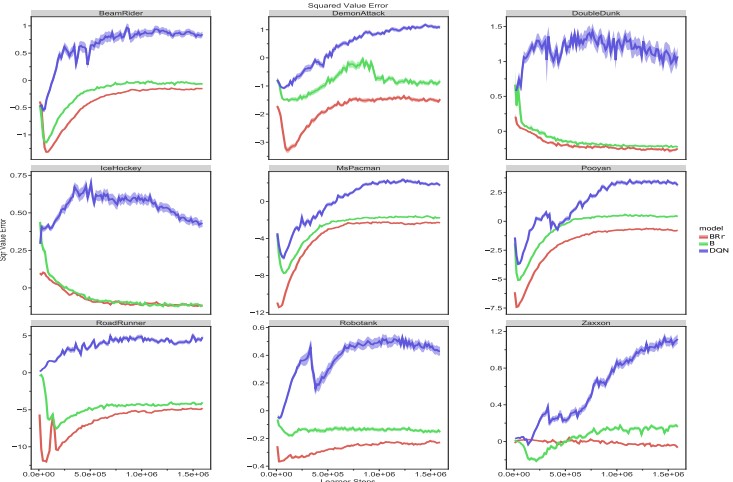

Figure 14: The squared value error computed in the environment by evaluating the agent and computed with respect to the ground truth discounted returns and reporting the mean squared values of the values.

## C  DETAILS OF DATASETS

### C.1  BSUITE DATASET

BSuite (Osband et al., 2019) data was collected by training DQN agents (Mnih et al., 2015) with the default setting in Acme (Hoffman et al., 2020) from scratch in each of the three tasks: cartpole, catch, and mountain_car. We convert the originally deterministic environments into stochastic ones by randomly replacing the agent action with a uniformly sampled action with a probability of $\epsilon \in \{0, 0.1, 0.2, 0.3, 0.4, 0.5\}$ (ie. $\epsilon = 0$ corresponds to the original environment). We train agents (separately for each randomness level and 5 seeds, i.e. 25 agents per game) for 1000, 2000, 500 episodes in cartpole, catch and mountain_car respectively. The number of episodes is chosen so that agents in all levels can reach their best performance. We record all the experience generated through the training process. Then to reduce the coverage of the datasets and make them more challenging we only used 10% of the data by subsampling it. More details of the dataset are provided in Table 2. The results presented in the paper are averaged over the 5 random seeds.

### C.2  DEEPMIND LAB DATASET

DeepMind Lab (Beattie et al., 2016) data was collected by training distributed R2D2 (Kapturowski et al., 2019) agents from scratch on individual tasks. First, we tuned the hyperparameters of a distributed version of the Acme (Hoffman et al., 2020) R2D2 agent independently for every task to achieve fast learning in terms of actor steps. Then, we recorded the experience across all actors during entire training runs a few times for every task. Training was stopped after there was no further progress in learning across all runs, with a resulting number of steps for each run between 50 million for the easiest task (`seekavoid_arena_01`) and 200 million for some of the hard tasks. Finally we built a separate offline RL dataset for every run and every task. See more details about these datasets in Table 3.

Additionally, for the `seekavoid_arena_01` task we ran two fully trained snapshots of our R2D2 agents on the environment with different levels of noise ($\epsilon = 0, 0.01, 0.1, 0.25$ for $\epsilon$-greedy action selection). We recorded all interactions with the environment and generated a different offline RL dataset containing 10 million actor steps for every agent and every value of $\epsilon$.

| Environments | Number of episodes | Number of transitions | Average episode length |
|---|---|---|---|
| cartpole ($\epsilon = 0.0$) | 1000 | 710K | 710 |
| cartpole ($\epsilon = 0.1$) | 1000 | 773K | 773 |
| cartpole ($\epsilon = 0.2$) | 1000 | 649K | 649 |
| cartpole ($\epsilon = 0.3$) | 1000 | 607K | 607 |
| cartpole ($\epsilon = 0.4$) | 1000 | 672K | 672 |
| cartpole ($\epsilon = 0.5$) | 1000 | 643K | 643 |
| catch ($\epsilon = 0.0$) | 200 | 1.8K | 9 |
| catch ($\epsilon = 0.1$) | 200 | 1.8K | 9 |
| catch ($\epsilon = 0.2$) | 200 | 1.8K | 9 |
| catch ($\epsilon = 0.3$) | 200 | 1.8K | 9 |
| catch ($\epsilon = 0.4$) | 200 | 1.8K | 9 |
| catch ($\epsilon = 0.5$) | 200 | 1.8K | 9 |
| mountain_car ($\epsilon = 0.0$) | 50 | 10K | 205 |
| mountain_car ($\epsilon = 0.1$) | 50 | 10K | 210 |
| mountain_car ($\epsilon = 0.2$) | 50 | 22K | 447 |
| mountain_car ($\epsilon = 0.3$) | 50 | 13K | 277 |
| mountain_car ($\epsilon = 0.4$) | 50 | 12K | 250 |
| mountain_car ($\epsilon = 0.5$) | 50 | 24K | 494 |

Table 2: BSuite dataset details.

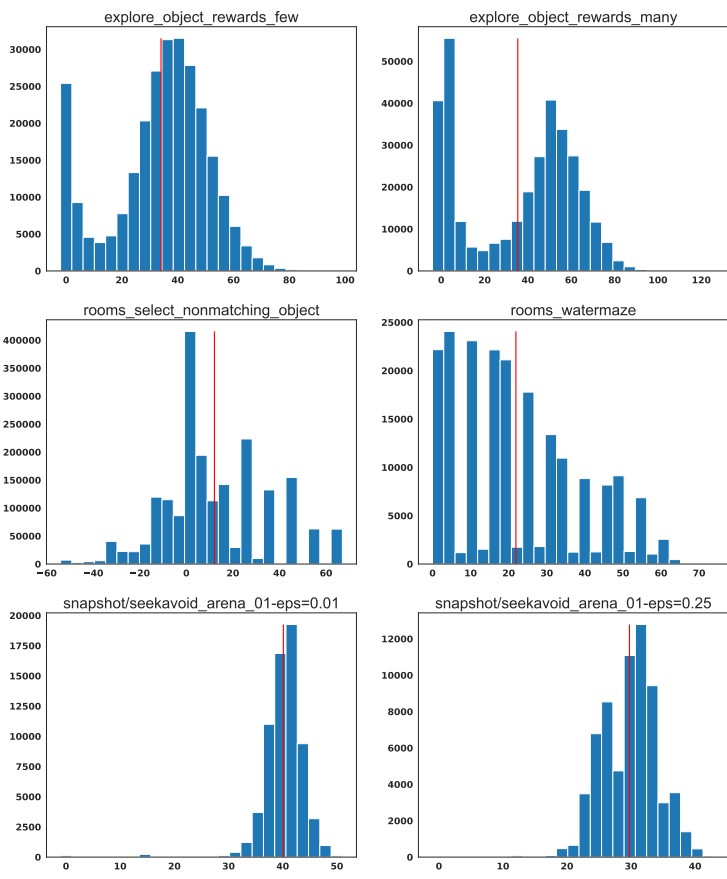

Figure 15: **DeepMind Lab Reward Distribution:** We show the reward distributions for the DeepMind Lab datasets. The vertical red line indicates the average episodic return in the datasets.

Table 3: **DeepMind Lab dataset details.** For training data, reward is measured as the maximum over training of the average reward over runs for the same task. For snapshot data, reward is just an average over all episodes recorded using the same level of noise.

| Task | Episode Length | Datasets | Episodes (K) | Steps (M) | Reward |
|------|----------------|----------|--------------|-----------|--------|
| `seekavoid_arena_01` | 300 | 5 | 667.1 | 200.1 | 39.0 |
| `seekavoid_arena_01` snapshot ($\epsilon = 0$) | 300 | 2 | 66.7 | 20 | 40.4 |
| `seekavoid_arena_01` snapshot ($\epsilon = 0.01$ | 300 | 2 | 66.7 | 20 | 40.1 |
| `seekavoid_arena_01` snapshot ($\epsilon = 0.1$) | 300 | 2 | 66.7 | 20 | 36.9 |
| `seekavoid_arena_01` snapshot ($\epsilon = 0.25$) | 300 | 2 | 66.7 | 20 | 29.7 |
| `explore_object_rewards_few` | 1350 | 3 | 178.3 | 240.7 | 51.5 |
| `explore_object_rewards_many` | 1800 | 3 | 334.1 | 601.4 | 64.5 |
| `rooms_select_nonmatching_object` | 180 | 3 | 2001.1 | 360.2 | 32.5 |
| `rooms_watermaze` | 1800 | 3 | 201.8 | 363.3 | 48.8 |

# D    EXPERIMENT DETAILS

We used the Adam optimizer (Kingma & Ba, 2014) for all our experiments. For details on the used hyperparameters, refer to the Table 4 for `bsuite`, Table 5 for Atari, and Table 6 for DeepMind Lab. Our evaluation protocol is described below, in Section D.1. On Atari experiments, we have normalized the agents' scores as described in (Gulcehre et al., 2020).

On Atari, in all our experiments we report the median normalized score along with the bootstraps estimates of 75th and 25th percentiles for the interquantile range estimates of the errors in the error bars as done by (Gulcehre et al., 2020).

**Atari Hyperparameters:**    On Atari we directly used the baselines and the hyperparameters reported in (Gulcehre et al., 2020), to get the detailed Atari results on test set we communicated with the authors. We have run additional CQL and our own models with ranking regularization and reparameterization. For CQL we have finetuned both the learning rate from the grid $[8e-5,\ 1e-4,\ 3e-4]$ and the regularization hyperparameter $\alpha \in [0.005, 0.05, 0.01, 0.1, 1]$. For our own proposed models we have only tuned the learning rate from the grid $[8e-5,\ 1e-4,\ 3e-4]$ and the ranking regularization hyperparameter from the grid $[0.005, 0.05, 0.01, 0.1, 1]$. We have fixed the rest of the hyperparameters. As mentioned earlier, we have only used the online policy selection games for finetuning the hyperparameters. As a result of our grid search, we have used learning rate of $1e-4$ for CQL and our models. We have used $0.01$ for the $\alpha$ hyperparameter of CQL. $0.05$ seems to be the optimal hyperparameter choice for the ranking regularization hyperparameter.

**DeepMind Lab Hyperparameters:**    On DeepMind Lab experiments, we tuned the hyperparameters of each model individually on each level separately. We have tuned the learning rate and the regularization hyperparameters for each model from the same grid that we have used for Atari. All our algorithms are n-step in DeepMind Lab experiments, where $n$ is fixed to 5 in all our experiments. Thus both behavior value estimation and Q-learning experiments use 5 steps of unrolls for learning.

## D.1    EVALUATION PROTOCOL

To evaluate the performance of the various methods, we use the following protocol:

1. We sweep over a small (5-10) sets of hyperparameter values for each of the methods.

2. We independently train each of the models on 5 datasets generated by running the behavior policy with 5 different seeds (ie. producing 25-50 runs per problem setting and method).

3. We evaluate the produced models in the original environments (without the noise).

4. We average the results over seeds and report the results of the best hyperparameter for each method.

### D.1.1 EVALUATION METHOD

To evaluate models (step 3. above), in the case of `bsuite` and DeepMind Lab we ran an evaluation job in parallel to the training one. It repeatedly read the learner's checkpoint and produced evaluation results during training. We report the average of the evaluation scores over the last 100 learning steps.

In the case of the Atari environments, instead of averaging performance during the final steps of learning, we take the final snapshot produced by a given method and evaluate it on a '100' environment steps after the training finished.

Table 4: **`bsuite` experiments' hyperparameters.** The top section of the table corresponds to the shared hyperparameters of the offline RL methods and the bottom section of the table contrasts the hyperparameters of Online vs Offline DQN.

| Hyperparameter | setting (for both variations) |
|---|---|
| Discount factor | 0.99 |
| Mini-batch size | 128 |
| Target network update period | every 2500 updates |
| Evaluation $\epsilon$ | $0.4^8$ |
| $Q$-network: | an MLP |
| $Q$-network: hidden units | $56, 56, \text{num\_actions}$ |
| Training Steps | 2M learning steps |
| Hardware | Tesla V100 GPU |
| Replay Scheme | Uniform |

| Hyperparameter | Online | Offline |
|---|---|---|
| Min replay size for sampling | 20,000 | - |
| Training $\epsilon$ (for $\epsilon$-greedy exploration) | 0.01 | - |
| $\epsilon$-decay schedule | 250K steps | - |
| Fixed Replay Memory | No | Yes |
| Replay Memory size | 1M steps | 2M steps |
| Double DQN | No | Yes |

Table 5: **Atari experiments' hyperparameters.** The top section of the table corresponds to the shared hyperparameters of the offline RL methods and the bottom section of the table contrasts the hyperparameters of Online vs Offline DQN.

| Hyperparameter | setting (for both variations) |
|---|---|
| Discount factor | 0.99 |
| Mini-batch size | 256 |
| Target network update period | every 2500 updates |
| Evaluation $\epsilon$ | $0.4^8$ |
| $Q$-network: channels | 32, 64, 64 |
| $Q$-network: filter size | $8 \times 8, 4 \times 4, 3 \times 3$ |
| $Q$-network: stride | 4, 2, 1 |
| $Q$-network: hidden units | 512 |
| Training Steps | 2M learning steps |
| Hardware | Tesla V100 GPU |
| Replay Scheme | Uniform |

| Hyperparameter | Online | Offline |
|---|---|---|
| Min replay size for sampling | 20,000 | - |
| Training $\epsilon$ (for $\epsilon$-greedy exploration) | 0.01 | - |
| $\epsilon$-decay schedule | 250K steps | - |
| Fixed Replay Memory | No | Yes |
| Replay Memory size | 1M steps | 2M steps |
| Double DQN | No | Yes |

Table 6: **Deepmind Lab experiments' hyperparameters.** The top section of the table corresponds to the shared hyperparameters of the offline RL methods and the bottom section of the table contrasts the hyperparameters of Online vs Offline DQN.

| Hyperparameter | setting (for both variations) | |
| --- | --- | --- |
| Discount factor | 0.997 | |
| Target network update period | every 400 updates | |
| Evaluation $\epsilon$ | $0.4^8$ | |
| Importance sampling exponent | 0.6 | |
| Architecture | Canonical R2D2 (Kapturowski et al., 2019) | |
| Hyperparameter | Online | Offline |
| Hardware | 4x TPUv2 | 4x Tesla V100 GPU |
| Training Steps | 50-200M actor steps | 50K learning steps |
| Sequence Length | 120 (40 burn-in) | Full episode |
| Mini-batch size | 32 | 8 |
| Training $\epsilon$ (for $\epsilon$-greedy exploration) | $0.4, ..., 0.4^8$ | - |
| Replay Scheme | Prioritized (exponent 0.9) | - |
| Min replay size for sampling | 600K steps | - |
| Replay Memory size | 12M steps | 50-200M steps |

## D.2 ATARI OFFLINE POLICY SELECTION RESULTS

In Table 7, we show the performance of our baselines on different Atari Offline Policy selection games. We show that `QRr` outperforms other approaches significantly.

Table 7: **Atari Offline Policy Selection Results**: In this table, we list the median normalized performance of different baselines.

| Name | Normalized Score |
| --- | --- |
| BC | 50.8 |
| DDQN | 83.1 |
| CQL | 98.9 |
| BCQ | 102.6 |
| IQN | 104.8 |
| REM | 104.7 |
| QRr | **108.2** |

## D.3 DEEPMIND LAB DETAILED RESULTS

In Table 8, we have shown the results on the Deepmind Lab datasets. It is possible see from these numerica results that `BR` outperforms other approaches and `B` is still very competitive.

## D.4 BSUITE DETAILED RESULTS

We generated datasets and performed experiments analogous to these in Section 4.1 for *mountain_car* environment. We present results for all three environments in Table 9. `BRr` outperforms all the baselines.

## E REPARAMETRIZING THE Q-NETWORK

In all reparameterized critic experiments we have used the $\tanh(\cdot)$ activation function with refine gates to help with optimization (Gu et al., 2019). We have not tuned the hyperparameters of the

Table 8: **Detailed Results on the DeepMind Lab:** We provide the detailed results for each DeepMind levels along with the standard deviations.

|  | BC | R2D2 | CQL | B | BR |
|---|---|---|---|---|---|
| `explore_object_rewards_few` | $1.8 \pm 1.0$ | $19.8 \pm 4.0$ | $23.8 \pm 5.1$ | $23.7 \pm 3.8$ | $\mathbf{28.6} \pm 1.7$ |
| `explore_object_rewards_many` | $2.9 \pm 1.4$ | $8.5 \pm 3.4$ | $9.3 \pm 2.5$ | $7.6 \pm 3.1$ | $\mathbf{13.4} \pm 11.8$ |
| `rooms_watermaze` | $0.1 \pm 0.1$ | $2.7 \pm 1.4$ | $4.0 \pm 3.7$ | $9.9 \pm 2.7$ | $11.2 \pm 4.2$ |
| `rooms_select_nonmatching_object` | $1.1 \pm 4.6$ | $5.4 \pm 2.3$ | $3.4 \pm 2.4$ | $9.4 \pm 6.3$ | $\mathbf{10.4} \pm 9.6$ |
| `seekavoid_arena_01`, $\epsilon = 0$ | $28.02 \pm 7.6$ | $4.7 \pm 3.0$ | $12.8 \pm 10.7$ | $4.4 \pm 0.9$ | $17.07 \pm 10.1$ |
| `seekavoid_arena_01`, $\epsilon = 0.01$ | $33.0 \pm 1.3$ | $5.5 \pm 1.6$ | $12.7 \pm 5.4$ | $4.1 \pm 1.8$ | $19.8 \pm 4.9$ |
| `seekavoid_arena_01`, $\epsilon = 0.1$ | $18.9 \pm 14.4$ | $8.6 \pm 3.0$ | $16.3 \pm 7.7$ | $11.775 \pm 4.5$ | $31.8 \pm 4.7$ |
| `seekavoid_arena_01`, $\epsilon = 0.25$ | $17.46 \pm 7.5$ | $13.5 \pm 5.1$ | $13.5 \pm 5.06$ | $9.0 \pm 0.25$ | $25.57 \pm 7.0$ |

| Environments | DDQN | BCQ | REM | CQL | BRr | QRr |
|---|---|---|---|---|---|---|
| cartpole ($\epsilon = 0.0$) | 203.5 | 244.3 | 383.7 | 354.6 | 933.8 | 358.3 |
| cartpole ($\epsilon = 0.1$) | 240.5 | 244.6 | 218.0 | 673.7 | 886.8 | 732.7 |
| cartpole ($\epsilon = 0.2$) | 134.5 | 215.7 | 295.6 | 528.3 | 786.1 | 566.0 |
| cartpole ($\epsilon = 0.3$) | 265.9 | 432.8 | 248.2 | 594.6 | 937.3 | 642.0 |
| cartpole ($\epsilon = 0.4$) | 278.9 | 418.4 | 263.8 | 791.3 | 814.5 | 745.2 |
| catch ($\epsilon = 0.0$) | -0.04 | 0.96 | 0.3 | 1.0 | 1.0 | 1.0 |
| catch ($\epsilon = 0.1$) | -0.19 | 0.85 | 0.18 | 1.0 | 1.0 | 1.0 |
| catch ($\epsilon = 0.2$) | 0.08 | 0.91 | 0.34 | 1.0 | 1.0 | 0.99 |
| catch ($\epsilon = 0.3$) | -0.08 | 0.92 | -0.05 | 1.0 | 0.99 | 1.0 |
| catch ($\epsilon = 0.4$) | -0.13 | 0.85 | 0.14 | 1.0 | 1.0 | 0.99 |
| mountain_car ($\epsilon = 0.0$) | -196.5 | -142.0 | -116.3 | -129.3 | -130.3 | -128.7 |
| mountain_car ($\epsilon = 0.1$) | -231.5 | -145.2 | -167.0 | -135.6 | -127.1 | -141.5 |
| mountain_car ($\epsilon = 0.2$) | -158.3 | -161.1 | -118.6 | -120.0 | -116.7 | -140.3 |
| mountain_car ($\epsilon = 0.3$) | -316.6 | -180.9 | -128.3 | -154.8 | -125.0 | -137.4 |
| mountain_car ($\epsilon = 0.4$) | -125.1 | -133.7 | -127.6 | -128.9 | -127.1 | -163.6 |

Table 9: BSuite mean results.

reparameterization in our experiments, we have used four times larger minibatches to update the scale, since it is cheap to update a single scalar and as shown in Algorithm 1, we have used twice smaller learning rate to update the scale than the rest of the parameters of the network. This is a heuristic, but we found this simple heuristic to work well across all the tasks that we have tried. Potentially it is possible to get better results by tuning the hyperparameters for reparameterization more carefully.

---

**Algorithm 1** Algorithm of Reparametrized Q-Network

---

**Inputs:** Dataset of trajectories $\mathcal{D}$, batch size to update $\theta$: $B1$, batch size to update $\gamma$: $B2$, and number of actors $A$.
Initialize $\hat{Q}$ weights $\theta$.
Initialize $\alpha$ to 1.
Initialize target policy weights $\theta' \leftarrow \theta$.
**for** $n_{\text{steps}}$ **do**
    Sample transition sequences $(s_{t:t+m}, a_{t:t+m}, r_{t:t+m})$ from dataset $\mathcal{D}$ to construct a mini-batch of size $B$.
    Calculate loss $\mathcal{L}(s_t, a_t, r_t, s_{t+1}; \theta, \alpha)$ using target network.
    Update $\theta$ with GD: $\theta \leftarrow \theta - \eta_1 \nabla_\theta \mathcal{L}(\theta)$
    Update $\alpha$ with GD: $\alpha \leftarrow \alpha - \eta_1 \sqrt{B_1/B2} \nabla_\gamma \mathcal{L}(\gamma)$
    If $t \mod t_{target} = 0$, update the target weights and $\alpha$, $\theta' \leftarrow \theta$, $\alpha' \leftarrow \alpha$.
**end for**

---

As seen in Algorithm 1, there is a two stage of optimization to update the parameters of Q-network $\theta$ and the scale of the Q values $\alpha$. They both use different learning rates, it is important to make sure that we update the $\alpha$ with a smaller learning rate: $\eta_2 \leq \eta_1$.

## F  Ranking Regularizer

We propose a family of methods that prevent the extrapolation error by suppressing the values of the actions that are not in the dataset. We achieve that by ranking the actions in the training set higher than the ones that are not in the training set. For the learned Q-function the absolute values of actions do not matter, we are rather interested in relative ranking of the actions. Given $a_t$ is the action from the dataset. For all $j \neq t$ and illustration purposes, the value iteration can be written as:

$$\mathbb{E}[\max_a Q(s,a)] \approx \mathbb{E}\left[P(Q(s,a_t) > Q(s,a_j))Q(s,a_t)|t \in Max\right] + \mathbb{E}\left[P(Q(s,a_t) \not> Q(s,a_j))Q(s,a_j)|j \in Max\right]$$

$$= \mathbb{E}\left[P(Q(s,a_t) > Q(s,a_j))Q(s,a_t)|t \in Max\right] + \mathbb{E}\left[(1 - P(Q(s,a_t) > Q(s,a_j)))Q(s,a_j)|j \in Max\right]$$

$$= \alpha\mathbb{E}\left[P(\hat{Q}(s,a_t) > \hat{Q}(s,a_j))\hat{Q}(s,a_t)|t \in Max\right] + \alpha\mathbb{E}\left[(1 - P(\hat{Q}(s,a_t) > \hat{Q}(s,a_j)))\hat{Q}(s,a_j)|j \in Max\right]$$

$$= \alpha\left(\mathbb{E}\left[P(\hat{Q}(s,a_t) > \hat{Q}(s,a_j))\hat{Q}(s,a_t)|t \in Max\right] + \mathbb{E}\left[(1 - P(\hat{Q}(s,a_t) > \hat{Q}(s,a_j)))\right]\right)\xi$$

where $\xi$ is an irreducible noise, because we can not gather additional data on $(s_t,\ a_j)$, and we don't know the corresponding reward for it. This causes extrapolation error which accumulates through the bootstrapping in the backups as noted by Kumar et al. (2019). We implicitly pull down the $P(Q(s,a_t) \not> Q(s,a_t))$ by ranking the actions in the dataset higher which pushes up $P(Q(s,a_t) > Q(s,a_j))$. As a result, the extrapolation error in Q-learning would also reduce.

### F.1  Pairwise Ranking Loss for Q-learning

In this section, we discus the relationship between the pairwise ranking loss for Q-learning and the list-wise pairwise ranking losses.

$$p_{tj} = \text{sigm}(\hat{Q})_\theta(s,a_t) - \hat{Q}_\theta(s,a_j))$$

$$\pi(a_t|s) \approx \prod_{i=0,i\neq t}^{|\mathcal{A}|} p_{ti}/Z$$

$$Z = \sum_{i=0}^{|\mathcal{A}|} \prod_{j=0,j\neq i}^{|\mathcal{A}|} p_{ij}$$

$$\mathcal{R}(\theta) = -\sum_{i=0}^{|\mathcal{A}|} \log(p_{ti})$$

$$= -\sum_{i=0}^{|\mathcal{A}|} \log(\text{sigm}(\hat{Q}_\theta(s,a_t) - \hat{Q}_\theta(s,a_j)))$$

$$= \sum_{i=0}^{|\mathcal{A}|} \text{softplus}(\hat{Q}_\theta(s,a_j) - \hat{Q}_\theta(s,a_t))$$

We use a common approximation (Chen et al., 2009; Burges et al., 2005) to the softplus-based log-likelihood is to use a hinge-loss which can be seen as an approximation:

$$\mathcal{C}(\theta) = \sum_{i=0,i\neq t}^{|\mathcal{A}|} \max\left(\hat{Q}_\theta(s,a) - \hat{Q}_\theta(s,a_t) + \nu, 0\right)^2 \tag{13}$$

Imposing the constraint in Equation (13) can be harmful if the dataset has lots of suboptimal trajectories. Because this constraint will try to maximize the values of suboptimal actions in the

dataset. As a result, similar to Wang et al. (2020), we propose a filtering function to impose that constraints only on rewarding transitions:

$$\mathcal{C}(\theta) = \exp(G^{\mathcal{B}}(s) - \mathbb{E}_{s\sim\mathcal{D}}[G^{\mathcal{B}}(s)]) \sum_{i=0,i\neq t}^{|\mathcal{A}|} \max\left(\hat{Q}_\theta(s,a_i) - \hat{Q}_\theta(s,a_t) + \nu, 0\right)^2 \qquad (14)$$

### F.2 Relationship to the Policy Gradients

It is possible to drive the foirmulation that we use for the ranking regularizer from the policy gradient theorem to show the relationship. The Ranking Policy Gradient Theorem formulates the optimization of long-term reward using a ranking objective as done in Lin & Zhou (2020). The proof below illustrates the formulation process. Let us note that we apply the ranking regularization on the offline and off-policy data, such that thee formalism below only works when the behavior policy and target policy are equivalent, when the transitions are coming from on-policy data. If the ranking regularizer is used on the on-policy data it approximates the policy gradients, but it will not on the off-policy data.

Our construction is based on direct policy differentiation (Peters & Schaal, 2008; Williams, 1992) where the objective function is to $\theta^* = \arg\max_\theta J(\theta)$.

$$\nabla_\theta J(\theta) = \nabla_\theta \sum_\tau p_\theta(\tau) G^{\mathcal{B}}(s) \qquad (15)$$

$$= \sum_\tau p_\theta(\tau) \nabla_\theta \log p_\theta(\tau) G^{\mathcal{B}}(s)$$

$$= \sum_\tau p_\theta(\tau) \nabla_\theta \log \left(p(s_0)\Pi_{t=1}^T \pi_\theta(a_t|s_t)p(s_{t+1}|s_t, a_t)\right) G^{\mathcal{B}}(s)$$

$$= \sum_\tau p_\theta(\tau) \sum_{t=1}^T \nabla_\theta \log \pi_\theta(a_t|s_t) G^{\mathcal{B}}(s)$$

$$= \mathbb{E}_{\tau\sim\pi_\theta} \left[\sum_{t=1}^T \nabla_\theta \log \pi_\theta(a_t|s_t) G^{\mathcal{B}}(s)\right]$$

$$= \mathbb{E}_{\tau\sim\pi_\theta} \left[\sum_{t=1}^T \nabla_\theta \log \left(\prod_{j=1,j\neq i}^m p_{ij}\right) G^{\mathcal{B}}(s)\right]$$

$$= \mathbb{E}_{\tau\sim\pi_\theta} \left[\sum_{t=1}^T \nabla_\theta \sum_{j=1,j\neq i}^m \log\left(\text{sigm}(p_{ij})\right) G^{\mathcal{B}}(s)\right] \qquad (16)$$

$$= -\mathbb{E}_{\tau\sim\pi_\theta} \left[\sum_{t=1}^T \nabla_\theta \sum_{j=1,j\neq i}^m \text{softplus}(p_{ji}) G^{\mathcal{B}}(s)\right]$$

$$\approx -\mathbb{E}_{\tau\sim\pi_\theta} \left[\sum_{t=1}^T \nabla_\theta \left(\sum_{j=1,j\neq i}^m \text{rectifier}\left(Q(s,a_i) - Q(s,a_j)\right)\right) G^{\mathcal{B}}(s)\right], \qquad (17)$$

with baseline $\mathbb{E}_{s\sim\mathcal{D}}[G^{\mathcal{B}}(s)]$ it will be,

$$\approx -\mathbb{E}_{\tau\sim\pi_\theta} \left[\sum_{t=1}^T \nabla_\theta \left(\sum_{j=1,j\neq i}^m \text{rectifier}\left(Q(s,a_i) - Q(s,a_j)\right)\right) \left(G^{\mathcal{B}}(s) - \mathbb{E}_{s\sim\mathcal{D}}[G^{\mathcal{B}}(s)]\right)\right]$$
$$(18)$$

Then we apply the $\exp(\cdot)$ transformation on $(G^{\mathcal{B}}(s) - \mathbb{E}_{s\sim\mathcal{D}}[G^{\mathcal{B}}(s)]$ to impose this loss loss mostly on the rewarding trajectories, and we can turn the maximization problem to a minimization one with a flip of sign:

$$= \mathbb{E}_{\tau\sim\pi_\theta} \left[\sum_{t=1}^T \nabla_\theta \left(\sum_{j=1,j\neq i}^m \text{rectifier}\left(Q(s,a_i) - Q(s,a_j)\right)\right) \exp\left(G^{\mathcal{B}}(s) - \mathbb{E}_{s\sim\mathcal{D}}[G^{\mathcal{B}}(s)]\right)\right]$$
$$(19)$$

where the trajectory is a series of state-action pairs from $t = 1, ..., T$, i.e. $\tau = (s_1, a_1, s_2, a_2, ..., s_T)$. The gradients in (19) is exactly the gradients of the ranking regularizer.

