# OpenReview forum: "Addressing Extrapolation Error in Deep Offline Reinforcement Learning"
_ICLR.cc/2021/Conference — Reject_

### Official Review · AnonReviewer4 · 2020-10-19
**no sound knowledge to rely on**

**Rating:** 3
**Confidence:** 4

**Review:**

Summary:
The paper deals with offline aka batch RL for discrete actions. Three techniques ((i) behavior value estimation, (ii) ranking regularization, and (iii) reparametrization of the value function), which can be combined with each other, are presented. These techniques are compared with other methods in different experiments. Furthermore a new benchmark is being introduced.  It is claimed that in this new benchmark, the new techniques outperform state-of-the-art methods. Furthermore it is claimed that the presented method „behavior value estimation“, although it is only a one-step greedy optimization is typically already sufficient for dramatic gains.

Strong points:
The abstract and the first part of the introduction (the first 1.5 pages) are very well written and the problems of offline RL are very well presented. Also very good is the consideration that the existence of a behavior policy is a restriction that does not apply to every given dataset, as expressed in the terms "behavior policy(s)" and "coherent policy". However, it is not specified in the text what exactly is meant by "coherent policy".

Weak points:
The representation becomes increasingly unclear from page 2 onwards. None of the three techniques presented is sufficiently discussed and sufficiently tested. None of the statements is supported convincingly, although the paper already makes extensive use of references to the Appendix. There are 14 references in the main text to the Appendix and four to figures in the Appendix.

Recommendation:
In its current form, the experimental part of the paper is immature. A uniform structure is missing. The statements are not sufficiently substantiated. Therefore I recommend to reject the paper. It seems that there is not enough space to present and sufficiently verify all three techniques.

The claim "this one step is typically sufficient for dramatic gains as we show in our experiments (see for example Fig. 9)" is not sufficiently substantiated, because one cannot speak of "typically", as only "Atari online policy selection games" are considered. And additionally Fig. 9 is located in the Appendix.
The meaning of the error bars in Fig. 3 and Fig. 7 is not explained.
The (on first sight) counterintuitive result that the performance of CQL and QRr at cart-pole is higher at 40% noise than at 0% must be explained in the text or caption.
Because the presented ideas are not sufficiently examined and supported, no sound knowledge is generated on which the reader can rely.

Questions:
What is the meaning of error bars in Fig. 3 and Fig. 7?
What is meant by "BC"?

Additional feedback with the aim to improve the paper:
The abbreviation BC is not introduced. It is unclear what is meant by it.
It is unclear what is meant by "coherent policy“.
What is meant by "discrete offline RL algorithms“?
it is not clearly described that discrete actions are required.
In the sum, i runs from 0 to 100 and is divided by 100, but from 0 to 100 we count 101 (unfortunately there are no line numbers in the manuscript, to locate the sum).
"5e - 2" does not look nice, better is 0.05 or $5 \cdot 10^{-2}$.
In Figure 8, the measurement results should not be connected by lines. Lines should only be used for fits or predictions of theory.
In Appendix F the text width is not respected.
Please do not use \pm for the standard deviation, but for the specification of the uncertainty (aka error of the measurement) e.g. the standard error.

-----------------------------------------
(Dec 3.) Although I appreciate the feedback, my assessment of the paper remains unchanged.

---

> ### Author Response · Authors · 2020-11-17
> **About Typos and Experiments**
>
> > In its current form, the experimental part of the paper is immature. A uniform structure is missing. The statements are not sufficiently substantiated...
>
> We found that combining these three methods (BRr) is required to be used together to achieve the best results, as we showed in our experiments. That is why we decided to include all of them in this paper. We wanted to develop a general recipe that uses the methods introduced in our paper to achieve the best results for a given problem. For example, if the dataset is low-coverage, we suggested using BVE. The ranking loss seems to improve across all the datasets, and the reparameterization trick helps with the stability issues during the training (especially with Q-learning.)
>
> > The claim "this one step is typically sufficient for dramatic gains as we show in our experiments (see for example Fig. 9)" is not sufficiently substantiated, because one cannot speak of "typically", as only "Atari online policy selection games" are considered...
>
> We only used the Atari online policy selections games for ablations and hyperparameter search. Figure 9 is showing the robustness of the model to the reward distribution and the dataset size on on-policy selection games. However, we provide the results on offline-policy selection games in Figure 4. We will fix that in the paper to refer to Figure 4 in that sentence.
>
> >  Questions: What is the meaning of error bars in Fig. 3 and Fig. 7? What is meant by "BC"?
>
> We use BC to mean "behavioral cloning", similarly to previous literature [1]. It is a very common acronym in reinforcement and imitation learning literature. We will define this acronym in the paper explicitly.
>
> [1] Torabi, Faraz, Garrett Warnell, and Peter Stone. "Behavioral cloning from observation." arXiv preprint arXiv:1805.01954 (2018).
>
> > What is meant by "discrete offline RL algorithms“? it is not clearly described that discrete actions are required.
>
> By "discrete offline RL algorithms“ we refer to offline RL algorithms with discrete actions.
>
> >  In the sum, i runs from 0 to 100 and is divided by 100…
>
> Thanks we will fix these typos.
>
> > In Figure 8, the measurement results should not be connected by lines. Lines should only be used for fits or predictions of theory...
>
> We will incorporate those changes suggested by the reviewer to the appendix.
>
> > Please do not use \pm for the standard deviation, but for the specification of the uncertainty (aka error of the measurement) e.g. the standard error.
>
> Thanks we will change that to the standard error.

---

### Official Review · AnonReviewer2 · 2020-10-26
**Promising experimental results but the algorithm has some limitations**

**Rating:** 4
**Confidence:** 4

**Review:**

##### Summary & recommendation
The paper proposes an offline RL algorithm, which consists of three techniques: behavior value estimation, ranking regularization, and reparametrization of Q function, to reduce overestimation errors. The algorithm is evaluated on several benchmark datasets.

Overall, the paper is clearly written. The empirical results also seem promising. However, my main concern is that the proposed algorithm, especially with the behavior value estimation technique, has a big limitation for solving the offline RL problem (details come below). Moreover, I don’t think the paper provide enough justification for using all three techniques other than experiment results. It would have been better if the authors could discuss why we need all these three techniques (e.g. maybe behavior value estimation + ranking regularization is similar to behavior regularization?), rather than just combing these tricks to make the algorithm work empirically. I think there is still room for improvement before publishing this paper. Therefore, I recommend to reject the paper.

##### Supporting arguments
The goal of an offline RL algorithm is to find a nearly optimal policy $\pi$ from an offline dataset. However, the behavior value estimation technique is just learning the value function for the behavior policy. Even though we perform a single policy improvement step in the test time, it is generally not sufficient to obtain a nearly-optimal policy, especially when the behavior policy is far from optimal.

The authors mention “Fortunately, this one step is typically sufficient for dramatic gains as we show in our experiments (see for example Fig. 9). This finding matches our understanding that policy iteration algorithms typically do not require more than a few steps to converge to the optimal policy (Lagoudakis & Parr, 2003; Sutton & Barto, 2018, Chapter 4.3)”. I think this is not true. Even in tabular case, policy iteration requires a polynomial time w.r.t. the size of the state space and the action space (e.g., see Theorem 1.14 of [1]). I think it is possible to construct a family of MDPs and some behavior policies such that one step of policy improvement is not sufficient. In such case, I think the proposed algorithm would not work well.

The related work section should not just list previous works, but explain how the proposed algorithm is different from or similar to the existing algorithms.

I have some questions to clarify my understanding of the paper:
* I am not sure what is the propose of Appendix A? Maybe I missed some important points here, but it has already been shown that off-policy + function approximation + bootstrapping can diverge (Sutton and Barto’s book).
* What is the exact definition of extrapolation error used in this paper? The paper mentions extrapolation over-estimation, extrapolation under-estimation, training time extrapolation error, and testing time extrapolation error, but I don’t see a clear definition of these terms.
* Regarding the experiments: How were $\nu$, $\beta$, and $\alpha$ selected? The algorithm introduces more hyper-parameters, so I wonder do you have any comments on hyperparameter search (e.g. do existing algorithms also require tuning extra hyper-parameters?). Do you have any reason why it needs a larger mini-batch and a smaller learning rate to update $\alpha$?

##### Minor comments
* $\theta’$ in Equation (2) is not defined

[1] Alekh Agarwal, Nan Jiang, Sham Kakade, and Wen Sun. Reinforcement Learning: Theory and Algorithms.

---

> ### Author Response · Authors · 2020-11-17
> **About the Quoted Theorem and Limitations**
>
> > ... The authors mention “Fortunately, this one step is typically sufficient for dramatic gains as we show in our experiments (see for example Fig. 9). This finding matches our understanding that policy iteration algorithms typically do not require more than a few steps to converge to the optimal policy (Lagoudakis & Parr, 2003; Sutton & Barto, 2018, Chapter 4.3)”. I think this is not true. Even in tabular case, policy iteration requires a polynomial time w.r.t. the size of the state space and the action space (e.g., see Theorem 1.14 of [1]). I think it is possible to construct a family of MDPs and some behavior policies such that one step of policy improvement is not sufficient...
>
> While it is possible to construct an MDP where 1-step is not enough, so far on the datasets that we have tried, this does not seem to be an issue. In particular, the improvement obtained with one-step of policy improvement in low-coverage datasets provides quite significant gains compared to regular Q-learning in the same settings.
> We would like to point out that the theorem cited in [1], forms an upper bound to the number of steps needed in the worst case scenario. It proves that policy iteration needs at most $O(polynomial)$ steps to reach the optimal performance starting from any $\pi^{0}$. The theorem, therefore, supports our claims that policy iteration is efficient. With a decent $\pi^{0}$ policy, you should expect to converge quickly. In the case where you start from the optimal policy, the number of  improvement steps needed is actually 0.
>
> > The related work section should not just list previous works, but explain how the proposed algorithm is different from or similar to the existing algorithms.
>
> In the related work section, we highlight some of the most important papers published recently on offline RL, in general. Nevertheless, we talk about other closely related works contrasting to ours throughout the paper.
>
> > I have some questions to clarify my understanding of the paper: I am not sure what is the propose of Appendix A? Maybe I missed some important points here, but it has already been shown that off-policy + function approximation + bootstrapping can diverge (Sutton and Barto’s book).
>
> It is true that off-policiness with a form of function approximation and bootstrapping has been known to diverge. But it hasn’t been shown before that the divergence necessarily will also happen with the neural networks as well. In Appendix A we give an example of such a case and show an example of how a non-linear neural network’s q-value estimation can diverge which to best of our knowledge was not shown before.
>
> > What is the exact definition of extrapolation error used in this paper?...
>
> Thanks for your comment. We tried to define these terms and try to give references to the related literature in Section 2.1. We will try to make make them more clear in the paper.
>
> > Regarding the experiments: How were  ν,  β, and α selected? The algorithm introduces more hyper-parameters, so I wonder do you have any comments on hyperparameter search (e.g. do existing algorithms also require tuning extra hyper-parameters?). Do you have any reason why it needs a larger mini-batch and a smaller learning rate to update α ?
>
> For the hyperparameter tuning please see our response to AnonReviewer1. Yes our motivation was basically to reduce the variance or noise in the gradients that arises during the training with small minibatches. Since α is just a single scalar, the using larger minibatches comes with almost no extra cost.
>
> > Minor comments θ′  in Equation (2) is not defined
>
> Thanks for pointers, we will fix this in the paper.

---

> > ### Comment · AnonReviewer2 · 2020-11-19
> > **Short Reply**
> >
> > Thanks for the response.
> >
> > "With a decent $\pi^0$ policy, you should expect to converge quickly. "
> > That might be true. However, the claim of the paper would be different. If the paper only considers the situation where the behavior policy is good or one PI step is sufficient, the paper should (1) explicitly mention it and (2) compare to appropriate baselines for the setting (e.g., behavioral cloning or safe policy improvement from [1]). Another issue is that how can we know if one PI step is sufficient? On the other hand, if the paper claims that their method works in general offline RL setting (as mentioned in my original review), I would consider the behavior value estimation method has a big limitation.
> >
> > [1] Thomas, Philip, Georgios Theocharous, and Mohammad Ghavamzadeh. "High confidence policy improvement." In International Conference on Machine Learning. 2015.

---

> > > ### Author Response · Authors · 2020-11-20
> > > **On theory and the expectations to solve general offline RL for any MDP with any behavioral policy**
> > >
> > > Thanks for your quick reply. We don't claim that BVE doesn't have any limitations; we will make this clearer in the paper. Nevertheless, we stand by our claim that the theorem R2 quoted, being an upper bound, and the references we cited in the paper support our claim regarding the one-step policy improvement.
> > >
> > > We think that the reviewer expects that "to claim to be solving a general offline RL, one needs to be able to solve any MDP with any behavioral policy." However, we find this expectation unrealistic.
> > >
> > > For example, let's assume having a dataset generated by two MDPs, both with one state and two actions: the first action always gives a reward of 0 in both MDPs. In the first MDP, the second action gives a reward of +1; in the second MDP, the same action receives a reward of -1. On the other hand, the behavioral policy, in both cases, always chooses the first action. No offline RL algorithm can successfully solve both of these MDPs at the same time with any number of policy improvements.
> > > As such, having access to a reasonable behavior policy is an inherent part of the offline setting, not our method's limitation. This is also likely why no other published offline RL method gives the convergence guarantees for any MDP/behavioral policy as R2 seems to expect from us.
> > >
> > > Moreover, R2 also asks us to compare against BC. We already compare our methods against BC in our experiments. We report BC results on every dataset in the paper except bsuite because, unsurprisingly, BC performed very poorly on that dataset due to the noisy transitions and small dataset sizes. Instead, we decided to include  BCQ on bsuite which is a stronger offline RL baseline.

---

### Official Review · AnonReviewer1 · 2020-10-27
**Good empirical results but may be difficult to work in other domains**

**Rating:** 4
**Confidence:** 4

**Review:**

Summary:
This paper focuses on the problem of Q value over-estimation in offline reinforcement learning and proposes three approaches (tricks) to help solve this problem. (1) estimate Q value of behavior policy avoiding max-operator in Q learning and take greedy action according to the behavior value estimation. (2) introduce ranking loss to push down the value estimation of all unobserved state-action pairs to avoid over-estimation. (3) use tanh operator to bound the range of Q value estimation, and learn a scale parameter with regularization term. The experimental results on several domains (Atari, Bsuite, Deepmind Lab) with discrete action space show performance better than existing algorithms.

Clarity:
This paper is generally written clearly, though I have several questions about the technique and experiments, which may need more clarification. Please see 'Cons' part for the detailed questions.

Originality:
The techniques of ranking regularization and re-parameterization of Q-values are novel in the literature of offline reinforcement learning. Behavior value estimation removing the max operator to alleviate over-estimation seems not novel, because many previous work in offline RL (e.g. SPIBB, ABM, CRR) use bellman operator without max operature for policy evaluation of target policy. Also, the one-step policy improvement in section 3.1 seems not novel, many previous work (e.g. BEAR, BCQ) sample the action according to the learned policy and take one of the sampled action with maximum value estimation at test time in the implementation.

Significance:
The main concern of the proposed method is whether it is theoretically sound and performs well in the other domains (such as domains with continuous action space) without much tuning of hyper-parameters (weight of regularization term when combining the proposed approaches). The three tricks are intuitive and might be useful in practice, but I am not sure whether the contributions are significant enough to match the acceptance bar of ICLR.

Pros:
* This paper attempts to solve a significant problem (extrapolation error in offline RL).
* The paper explains the intuition behind each proposed approach clearly.
* The experimental results are good on several domains with discrete action space, better than the baseline methods.

Cons:
* In section 1, "Surprisingly, this technique with only one round of improvement ... often outperform existing offline RL algorithms" seems a bit misleading and overclaiming. The proposed technique (1) can be better than behavioral policy only when the behavior policy is not deterministic and greedy with respect to the value estimation. And the experiment only verifies that it can outperforms the existing methods in some specific datasets collected on domains with discrete action. I doubt whether this technique can "often outperform" the existing algorithms (e.g. ABM, CRR, CQL) on continuous control tasks.
* Overall, the proposed techniques (2) (3) are not quite convincing without the support of the theory. In equation (5), the specific formulation of the weight of regularization, e.g. $exp((G^B(s)−E_{s∼D}[G^B(s)])/β)$ is not well-motivated. Why we use exp function instead of another simpler monotonically increasing function? Why we need the coefficient β? How to choose the value of ν and β (the given value in the paper are just randomly picked)? In equation (7), the regularization for learning the scale parameter is not well-motivated either. Is it really necessary to have this regularization? Why square function here is better than other functions? Without a theoretical ground, all these design choices and value choices are just like heuristic or magic numbers. The experiments show these choices can work on some dataset (perhaps with much tweak and tuning), but we are not confident whether they can also work on new datasets.
*In section 4.1 and 4.2, the performance of the proposed method is not significantly better than the baselines (the error bar of the proposed method overlaps with the error bar of the baselines).
*In section 4.1 and 4.2, QRr and BRr are mainly considered, but in section 4.3, only B and BR are considered. I am curious whether BRr and QRr perform well in this domain? If not, could you please explain why different combinations of the three propose techniques perform differently in different domains? Is there any principle about which combination should be used in which kind of dataset?
*In Figure 10 and 11, it seems that on Atari games, the larger weight of the ranking regularization means better performance. Then why "0.05 seems to be the optimal choice for the ranking regularization hyper-parameter"? Is the hyper-parameter value 0.05 used across all the datasets? Is the proposed approach sensitive to this value?
*In Algorithm 1, is it a typo of using γ? In the main text, γ means the discounting factor, then what's the definition of $\mathcal{L}(\gamma)$? The details of the re-parameterization of Q network need more clarification.

---

> ### Author Response · Authors · 2020-11-17
> **On the novelty of BVE, Lack of Theory and Significance**
>
>  > Behavior value estimation removing the max operator to alleviate over-estimation seems not novel, because many previous work in offline RL (e.g. SPIBB, ABM, CRR)... Also, the one-step policy improvement in section 3.1 is not novel,...
>
> The reviewer states that Behavior Value Estimation (BVE) is not novel, because it is similar to behavior constrained offline RL methods, like SPIBB, ABM, CRR, BEAR, and BCQ. However, none of these methods attempt to estimate the value of the behavior policy directly, nor do they perform policy improvement in one step. Instead, all of them jointly estimate the value of a learned policy, and improve the policy according to that value estimate. (In BCQ this policy is implicit, but Q is the value of the learned policy $\pi$, and not the behavior policy $\pi_b$). This combination is prone to over-estimation. We bypass this issue with BVE.
>
> > Significance: The main concern of the proposed method is whether it is theoretically sound and performs well ...
>
> In this paper, we focused on discrete actions, and we showed extensive results on many discrete-action tasks across three different domains: bsuite, Atari, and Deepmind Lab. RL with discrete actions can be applied to real-world applications, and continuous control problems can be cast as a discrete control problem. Additionally, there have been other impactful works in offline RL which focused on discrete actions [1,2]. However, we believe that our method can be extended to continuous action domains.
>
> Hyper-parameter tuning is a typical process throughout DL and Deep RL literature. While we believe that our method is robust to hyper-parameters, this is not something any DL system can guarantee except through empirical evidence.
>
> [1] Fujimoto, Scott, Edoardo Conti, Mohammad Ghavamzadeh, and Joelle Pineau. "Benchmarking batch deep reinforcement learning algorithms." arXiv preprint arXiv:1910.01708 (2019).
>
> [2] Agarwal, Rishabh, Dale Schuurmans, and Mohammad Norouzi. "An Optimistic Perspective on Offline Reinforcement Learning." (2020).
>
> > … The proposed technique (1) can be better than behavioral policy only when the behavior policy is not deterministic ...
>
> There are two parts to answer this concern. The first one is regarding how likely it is for the behavior policy to be deterministic and greedy concerning its value estimation. In this paper, we focused on stochastic datasets. The datasets generated from real-world environments are rarely deterministic and greedy, mainly because the world is noisy, and greedy policies can not give good coverage to learn policies. Often demonstrations gathered from humans can be considered as stochastic too. Thus, we decided to focus on environments generated by stochastic policies.
>
> Now, in terms of evidence that one-step of policy improvement performs better, in Figure 8, we have run experiments to verify how well our proposed methods perform when compared to BC, R2D2, and CQL while changing the epsilon (noise level) if fixed behavior policy. In Figure 8, noise level 0 corresponds to a deterministic and a greedy policy. As shown in that figure, both BVE and R2D2 perform equally poorly in that setting. In contrast, behavior cloning, and BVE with ranking regularization perform much better.
>
> > Overall, the proposed techniques (2) (3) are not quite convincing without the support of the theory.
>
> Unfortunately, it is unclear for us what the reviewer means by theory in this comment. Is the question whether the updates will converge? Convergence proofs in general can not be provided when one deals with non-linear function approximators. Even ignoring the function approximator, most DRL systems employ additional terms to the loss (e.g. entropy regularization for actor-critic) which breaks any hope of ensuring that the updates will lead to a fixed point.
>
> E.g. For a tabular case, (2) will simply make the value of any action-state pair not in the dataset to be below the value of the action-state pairs within the datasets. One could potentially build on this to show that in the tabular case, (2) can not force learning to diverge from the best solution you can find given the data that you have (e.g. if you initialize under the optimal policy under the data that you have you will not move away from it). However such a proof will have minimal implications once we go to the large practical scale of things that involve neural networks. To what extent do such proofs (or theory) provide any extra confidence above the empirical evidence?
>
> (3) is guaranteed to prevent the divergence by bounding the critic, in some sense it is akin to vmin and vmax in distributional RL, however instead of setting the vmin and vmax as a hyperparameter, here we proposed to learn the scale of the Q-network’s outputs from the data.

---

> ### Author Response · Authors · 2020-11-17
> **On the Motivation, Design Choices and Hyperparameters**
>
> > In equation (5), the specific formulation of the weight of regularization, e.g.  Β in exp((GB(s)−Es∼D[GB(s)])/β) is not well-motivated. Why we use exp function instead of another simpler monotonically increasing function? Why we need the coefficient β?
>
> This formulation is based on the filtering mechanism proposed in the CRR paper. In CRR paper, they have used the advantage function from the critic to filter out the transitions. Moreover, CRR paper ablated and reported better results with exp function instead of for instance the indicator function. In contrast to CRR, in this paper, we rely on the discounted returns, which, according to our preliminary experiments, seem to be more reliable since we don’t have access to a policy directly.
>
> > How to choose the value of ν and β (the given value in the paper are just randomly picked)?
>
> We performed a small grid search (v in [0.005, 0.05, 0.5] and β in [2, 1, 0.5]) on a small number of Atari games. We used the best setting from this search for all other datasets and tasks.
>
> > In equation (7), the regularization for learning the scale parameter is not well-motivated either. Is it really necessary to have this regularization?
>
> Yes, in our preliminary experiments, we have verified that disabling the regularization causes a degradation in the performance. We can add this to the appendix, if the reviewers find this insight useful.
>
> > Why square function here is better than other functions?
>
> We don’t claim that squared hinge loss is better. Square hinge loss is very common in the learning to rank literature [3] due to some of its properties mostly for SVMs and kernel machines. However, with deep learning models we don’t think the choice between square hinge loss and non-squared one will make a big difference. We decided to use the squared hinge loss for two reasons: first, it was easier to establish its relationship to the other known RL methods such as ranking policy gradients, and second it punishes the small errors less and large errors more.
>
> [3] Chapelle, Olivier, and S. Sathiya Keerthi. "Efficient algorithms for ranking with SVMs." Information retrieval 13, no. 3 (2010): 201-215.
>
> > Without a theoretical ground, all these design choices and value choices are just like heuristic or magic numbers...
>
> We agree that a theoretical grounding might be helpful in terms of understanding, but we decided to go with more breadth with respect to environments and empirical analysis in this paper rather than providing novel theoretical insights. The theory is useful, but we need to believe that for offline RL literature we need more practical papers providing empirical insights on the datasets we have. We hope that our clarification above has clarified the reviewers misinterpretation.
>
> > In section 4.1 and 4.2, the performance of the proposed method is not significantly better than the baselines (the error bar of the proposed method overlaps with the error bar of the baselines).
>
> In Section 4.1, we used standard deviation for the plots, however, after we switched to standard errors and the error bars are better both in section 4.1 and 4.3. The errors reported in 4.2 are the error across the Atari games. We realized that all methods can have very high variances across the Atari games. The part of the reason is because some Atari games need to be run long and  are very large-scale, however, sometimes due to the cluster infrastructure, learners can be interrupted or can go down because of hw issues, which can introduce noise in the performance. Let us note that our baselines also have high noise in 4.2 as well, which we took from [4].
>
> [4] Gulcehre, Caglar, Ziyu Wang, Alexander Novikov, Tom Le Paine, Sergio Gómez Colmenarejo, Konrad Zolna, Rishabh Agarwal et al. "RL Unplugged: Benchmarks for offline reinforcement learning." arXiv preprint arXiv:2006.13888 (2020).

---

> ### Author Response · Authors · 2020-11-17
> **Hyperparameters and Typos**
>
> >... I am curious whether BRr and QRr perform well in this domain? If not, could you please explain why different combinations of the three propose techniques perform differently in different domains? Is there any principle about which combination should be used in which kind of dataset?
>
> In our Atari experiments, we noticed that the performance difference between QRr and BRr was not very significant. As we noted in the paper, we noticed that Behavior Value Estimation works very well if the coverage in the dataset is low (see Figure 9 for Atari coverage experiments). We also realized that *Behavior Value Estimation* significantly improves the performance on Deepmind Lab datasets over regular Q-learning (R2D2). Thus we decided to only focus on BR. We confirmed this by comparing QR and BR on the Deepmind Lab on the seekavoid dataset. We ran experiments with BRr as well, and the results were only slightly better than BR. We will add the QR and BRr results to the paper as well. Unlike QR, since BR doesn’t get affected by over-estimation, we realized the reparameterization trick provides a smaller improvement when used with BR.
>
> > In Figure 10 and 11, it seems that on Atari games, the larger weight of the ranking regularization means better performance. Then why "0.05 seems to be the optimal choice for the ranking regularization hyper-parameter"? Is the hyper-parameter value 0.05 used across all the datasets? Is the proposed approach sensitive to this value?
>
> Figure 10 just shows how the regularization hyperparameter affects the action gap and Figure 11 shows how that hyperparameter influences over-estimation. They don’t tell anything with respect to the performance. According to Figure 10, we can arrive at the conclusion that increasing the regularization coefficient can result in lower estimation error and better optimization but doesn’t necessarily mean that it will cause better performance as we discussed in the text. In Figure 11, we show the effect of increasing the regularization on the overestimation of the Q network when evaluated in the environment where the hyperparameter we used for atari (0.05) achieves lower over-estimation error.
>
> We used the hyperparameter value 0.05 for Atari and Deepmind  Lab datasets but on bsuite we realized using larger regularization values provide better performance for bsuite. In general, we would say relatively our method is quite robust to the regularization coefficient for the Atari, the best hyperparameters found on online-policy selection games (see Figure 5) performed very well on offline policy selection games (see Figure 4) where the hyperparameter search is not allowed.
>
> > In Algorithm 1, is it a typo of using γ? In the main text, γ means the discounting factor, then what's the definition of L(γ). The details of the re-parameterization of Q-network need more clarification.
>
> Yes, that is a typo, it was supposed to be alpha. We will fix this in the paper and clarify reparameterization of the Q-network.

---

### Author Response · Authors · 2020-11-17
**Clarifications to All Reviewers**

We would like to thank all the reviewers for their valuable comments and suggestions.

We would like to highlight that some of our reviewers have conflicting views on behavior value estimation (BVE):

* Reviewer 1 states that Behavior Value Estimation is not novel, because it is similar to behavior constrained offline RL methods, like SPIBB, ABM, CRR, BEAR, BCQ, CQL. However, none of these methods attempt to estimate the value of the behavior policy directly, nor do they perform policy improvement in one step. Instead, all of them jointly estimate the value of a learned policy, and improve the policy according to that value estimate. (In BCQ this policy is implicit, but Q is the value of the learned policy $\pi$, and not the behavior policy $\pi_b$). This combination is prone to over-estimation. And by using BVE, we bypass this issue.

* Reviewer 2 seems to disagree with Reviewer 1 on the novelty of BVE, but claims that it is not theoretically sound. R2 also cites a theorem, but as we discuss in our response to the reviewer, the theorem cited is a worst-case analysis and doesn’t make claims about the exact number of policy improvement steps needed given a policy in a practical setting.

Regarding the experiments and hyperparameters as raised by AnonReviewer1 and AnonReviewer2, beta and v are only tuned on Atari online policy selection games, and the best settings are used for the rest of the tasks. In addition, on Deepmind Lab and bsuite, we did a grid search for the regularization coefficient and learning rate.

We are going to fix the typos that are pointed out by the reviewers.

---

### Decision · Program_Chairs · 2021-01-07
**Final Decision**

**Decision:**

Reject

**Comment:**

This paper proposed a new method for improving offline RL. AC thinks that the paper has a potential, but all reviewers suggest rejection as the current write-up is quite poor. This causes many misunderstandings of reviewers. The authors clarify some misunderstandings/concerns in the discussion phase, but did not update the draft accordingly. Hence, AC cannot suggest acceptance, given the current form.